# Toward Food Security in 2050: Gene Pyramiding for Climate-Smart Rice

**Isnaini Isnaini [1,2], Yudhistira Nugraha [3], Niranjan Baisakh [4] and Nono Carsono [1,*]**

[1] Lab of Plant Breeding, Faculty of Agriculture, Universitas Padjadjaran, Sumedang 45363, Indonesia; isnaini@lecturer.unri.ac.id
[2] Faculty of Agriculture, Universitas Riau, Simpang Baru, Pekanbaru 28293, Indonesia
[3] National Research and Innovation Agency of Indonesia, Bogor 16911, Indonesia; yudh018@brin.go.id
[4] School of Plant, Environmental and Soil Sciences, Louisiana State University Agricultural Center, Baton Rouge, LA 70808, USA; nbaisakh@agcenter.lsu.edu
[*] Correspondence: n.carsono@unpad.ac.id

**Abstract:** The decline in crop productivity due to climate change is a major issue that threatens global food security and is the main challenge for breeders today in developing sustainable varieties with a wider tolerance to abiotic and biotic stresses. Breeding climate-smart rice (CSR) cultivars may be the best adaptation to climate change, with the potential to improve future food security and profitability for farmers in many nations. The main objective of this review is to highlight the direction of development of superior rice breeding from time to time, and various studies of new techniques of breeding methods for pyramiding various superior rice characteristics, especially characteristics related to abiotic stress, and to make a climate-suitable genotype that is resilient to climate change. For the design and strategy of the information search, a methodology was followed to compile and summarize the latest existing studies on rice breeding for abiotic stresses. The findings revealed that there is still an empty research gap in the context of supplying CSR products, which should be a priority for rice researchers in order to increase dissemination and ensure food security for future generations, particularly in climatically vulnerable agro-ecologies. And we conclude that, while technological innovation, specifically the integration of DNA markers and the genomic approach into conventional breeding programs, has made major contributions to the development of CSR, there is an urgent need to build strategic plans for the development of varieties with various stress tolerances.

**Keywords:** gene pyramiding; molecular-assisted selection (MAS); environment stress; climate-smart rice

## 1. Introduction

In the first two decades of the 21st century (2001–2020), the Earth's surface temperature became 0.99 °C warmer than it was in 1850–1900 and the global annual average is expected rise by another 0.85 °C within the next 100 years [1]. An increase in global temperatures will intensify heat waves, followed by wet and very dry weather. These causes changes in the pattern, intensity and frequency of rainfall, and unpredictable drought and flood conditions will occur more frequently. Therefore, it is critical to have adaptation technology to minimize the effect of the harsh environment on plant growth, development, and yield through genetic improvement.

Breeders have developed many high-yielding rice varieties with additional properties such as quality improvement and biotic and abiotic stress tolerance. Multi-tolerant rice has begun to be noticed by researchers through the combination of several resistance characteristics to biotic and/or abiotic stresses to address the challenges of future environmental conditions where an area frequently receives multiple stresses [2–7]. Multi-tolerance rice varieties can provide yield guarantees for farmers in areas that are frequently subjected to multiple types of stress in a single growing season. However, there is currently a very

limited variety with more than two additional tolerances/resistance traits available, particularly for the abiotic stress tolerance characteristic. This is due to the nature of abiotic stress tolerance that is inherited in a complex manner and phenotype selection methods require a large breeding material population and time consumption, which slows down selection for these characteristics.

Over the last three decades, climate change has triggered a significant trend in research to fine-tune the solution for agricultural problems caused by climate change [8]. More severe weather patterns such as drought, flooding/submergence, heat/cold, and soil problems such as salt and iron toxicity have had a large unfavorable impact on agriculture. There is an urgent need to develop a climate-adaptive rice genotype that can cope with climate change issues.

Climate-smart rice is a form of climate-smart agriculture in which rice varieties are developed to be able to withstand more intense harsh environments in the future [9]. A single area may experience several environmental stresses in a single growing season, at different stages of plant growth. It is expected that over the next few years, rainfed lowland areas will experience higher rainfall during the early stages of crop development resulting in flooding, followed by dry periods that will result in subsequent droughts. Under environmental stresses that frequently occur in one growing season, climate-smart rice is expected to help achieve a sustainable increase in rice production [10]. Climate-smart rice cultivars could have the potential to improve future food security while also increasing revenues for impoverished farmers in developing countries.

In this review, our main goal is to highlight the direction of superior rice breeding development within a specific time frame, and we discuss various studies of new breeding method techniques for a stacked multi-tolerance gene to abiotic stress into one superior rice variety. The objectives of this paper are also to update the latest research on the assembly of multi-tolerant rice which will also be summarized here, especially in completing research on the assembly of climate-smart rice.

## 2. Methodology

The research question in this paper was to understand the long journey of rice breeding trends that have been published in scientific articles. The researchers' concerns about the importance of climate-smart rice were extrapolated, as were method approaches that can be used to achieve this goal. The search themes were established to encompass as many aspects of rice breeding projects as was feasible. As a result, the peer-reviewed literature was searched for the terms "gene pyramiding", "rice breeding", "abiotic stresses in rice", "biotic stress in rice", "rice QTL", and "food security" from various scientific databases such as Springer, Sciencedirect, ResearchGate, Google Scholar, Oxford Academic, Wiley, and others. The electronic searching resulted in the documentation of 2891 articles. In order to eliminate typing errors and redundant references, the Mendeley tool was used to collect, manage, and create bibliographies. Through reading the abstracts and titles and eliminating duplicates, the search results were filtered.

This review paper compiled a variety of studies from the literature, primarily from 2013 and 2023, including research reports, reviews, academic articles, book chapters, books, and other editorial materials. The scientific articles used in this review were published more than ten years ago, accounting for a small portion of the collection.

Journals cited include those with the keywords of rice pyramiding, rice gene introgression and selection on abiotic stress, climate-smart agriculture, and climate-ready and climate smart rice. To ensure that no relevant articles were overlooked, thorough screening by full text reading followed by forward and backward snowball techniques were used.

A total of 238 documents, including research and review papers, policy assessments, and technical publications, were found through the search. The papers selected were distributed amongst the authors, and the relevant data were extracted through a literature review. The data from all the authors were compiled into a single document. After eliminating redundancy, the information was shared amongst the authors to finally modify,

add, or rearrange the data in a second brief review. After the completion of compilation and manuscript drafting, the JournalGuide search engine was used to explore the appropriate journal for publication.

## 3. Results

### 3.1. Rice Breeding Direction from Time to Time

Rice is the main crop and a staple food for more than half the world's population, more than 3.5 billion people worldwide, with Asia as the largest rice-producing and rice-consuming region [8]. Rice breeding programs continue to evolve dynamically, influenced by the need to increase production. In this section, the authors demonstrate how rice assembly technology has been used and progressed through several milestones, including before the Green Revolution, throughout the Green Revolution, after the Green Revolution, and during the cutting-edge age (Figure 1).

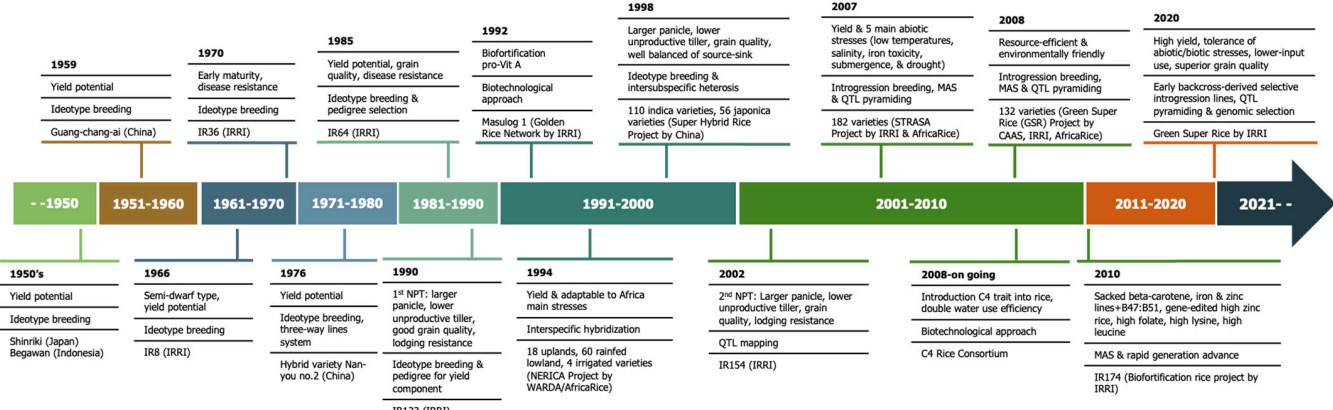

**Figure 1.** Milestones of the rice breeding program from time to time.

As a staple food, yield remains the primary selection criterion in developing outstanding rice varieties throughout history, as shown in Figure 1. Rice breeding during the Industrial Revolution until the Green Revolution was based on varieties selected for local adaptation by farmers with no understanding of genetic basis. The Shinriki (Rono) variety selected by Japanese farmers became a very famous variety in the 1890s and is known to have a shorter stature and is more responsive to nutrient inputs with higher yields [11]. Prior to the Green Revolution, rice variety development was primarily focused on assembling minimal input variants and those for less fertile rainfed land. Furthermore, it was created to develop fertilizer-responsive types. During this time, yield potential was defined as a variety's yield when grown under the suitable conditions to which it is adapted, with non-limiting nutrients and water and efficiently controlled pests, diseases, weeds, lodging, and other stressors [12].

According to reports, the first rice variety produced in Indonesia was var. Bengawan in 1943. The Bengawan type of rice has a genetic background of varieties from China, var. Latisail from India, and var. Benong from Indonesia [13]. Rice dwarf-type varieties have been widely regarded as one of the most significant breakthroughs in rice breeding history. Guang-chang-ai that was released in China in 1959 is the world's first semi-dwarf indica variety. The 85–100 cm height variety produced by crossing Ai-zai-zhan 4 with Guang-chang 13 was released on a large scale and initiated rice-dwarfing breeding all over the world [14]. This was followed by the release of Taichung (Native)-1 in Taiwan, derived from the spontaneous dwarf mutant Dee-Gee-Woo-Gen in 1949 [15].

Variety-IR8 was the first IRRI semi-dwarf, high-yielding modern rice variety for tropical irrigated lowlands, released in 1966. The dwarfing genes from Dee-geo-woo-gen were introduced into the Indonesian tall variety, PETA [16]. The Green Revolution influenced the direction of rice breeding programs in many countries, which were directed towards the advancement of high-yielding varieties (HYVs), which have a high yield

potential if grown in suitable environments, with adequate water and nutrients, and with other stresses such as weeds, pests, and diseases well managed. The adoption of IR 8 and its various derivatives is believed to mark the era of the Green Revolution in Asia. IR8 gradually became the base crop of the HYVs, yielding 10 tons ha$^{-1}$ with optimal crop management, which was 1–2 tons ha$^{-1}$ higher on irrigated lands compared with traditional varieties [17].

Around the same period in the 1960s, China successfully developed three lines for hybrid rice cultivation (cytoplasmic male sterile line, maintainer line, and restorer line). Twelve years later, the first F1 hybrid seeds were released for commercial production [18]. China became the first country to implement the large-scale production of hybrid rice technology [18]. The first hybrid rice varieties were three line hybrids with yields 15% to 20% higher than inbred or high-yielding varieties grown during the same growing period. It resulted in a significant rise in the amount of rice production in China, with more than 6 billion acres covered to date, boosting total rice production by more than 0.6 billion tons [19]. This technology has improved yield by approximately 2.5 million tons per year, with an expanded area to cover over 16 million ha per year, representing 57% of all rice cultivation and nearly 65% of total rice production over the last few decades. It is enough to feed an additional 80 million people and reach China's grain self-sufficiency goal, while also highlighting China's contribution to agricultural technology advancement.

Meanwhile, in South and Southeast Asia, IR36 was one of the most popular rice varieties in the early 1980s. With its resistance to disease and insects, this variety produces a higher yield in a shorter period of time, about 20 days less (111 DAP) than IR8 [20]. IR36 became the most grown rice at the time, covering more than 10 million acres. In 1985, IRRI developed the mega-variety IR64 [21]. Its characteristics include extensive adaptation, early maturity, excellent grain quality, short stature of plant height of approximately 100 cm, and resistance to brown planthopper biotype 3 compared with other previous IRRI HYVs. IR64 became an ideotype standard of high quality in the rice industry. IR64 replaced IR36 in the growing area of Asia, swiftly spreading to new areas. It is estimated that 10 million ha of IR64 was grown in the two decades after its release, with Indonesia as the major producer [22,23].

After the grand success of Green Revolution breakthroughs, rice yield has nearly hit stasis level. Since the 1990s, biotechnological inputs have been used to improve the selection procedure, which was previously based solely on visual phenotypic selection. Several success stories have been documented during the biotechnology era, including the golden rice project (IRRI), the China Super Rice Project (MoA of China), the Nerica Project (Warda/AfricaRice), and IRRI's biofortified rice.

One of the applications of genetic engineering is in golden rice. Golden rice was assembled in response to the vitamin A deficiency (VAD) condition, a severe public health problem that affects millions of children and pregnant women worldwide. Rice in its nature does not synthesize carotenoid chemicals in grains that humans consume, and this is combined with the absence of the necessary gene in the rice gene pool. This situation makes the rice endosperm lack provitamin A. Thus, the objective was to introduce the full metabolic pathway [24]. In 1992, a rice line containing the endosperm possessing high levels of provitamin A was introduced, which was delightfully apparent as a "golden" color of varying strength in different lines [25]. After an extensive safety assessment, the Philippines became the first country in the world to authorize golden rice for commercial multiplication in 2021 [26].

The West Africa Rice Development Association (WARDA) (now the AfricaRice Centre) initiated the New Rice for Africa (NERICA) project, which conducted interspecific hybridization between *Oryza glaberrima* (African rice) and *O. sativa* (Asian rice) to create a major scientific breakthrough in African rice development [27]. There were around 82 NERICA cultivars, including 18 upland, 60 lowland rainfed, and 4 irrigated varieties, that have been widely planted in more than ten Sub-Saharan African nations [28]. One of the most well-known cultivars today is NERICA-L-19, a rainfed lowland with higher

yields (5–7 t/ha), earlier maturation (in 75–100 days), and is largely adaptable to Africa's main stresses (drought and phosphate deficit), as well as being tolerant to iron toxicity and blast [29].

To address the food security of China's growing population, the Chinese Super Hybrid Rice project was launched, with the main goal of exceeding hybrid rice productivity targets 20 years earlier through better photosynthetic efficiency. This experiment incorporated two yield improvement methods: morphological improvement using the ideotype technique and inter-subspecific heterosis [30]. The ideotype targets are taller erect-leaf canopies for photosynthetic efficiency improvement, lower panicle position with larger panicle size for greater loading tolerance, and increased yield potential [30]. The super hybrid rice variety out-yielded ordinary hybrid rice by 30% with improvements in both source and sink [31].

It has been reported that modifying both physiological and morphological aspects could be feasible to increase yield potential by 25% [32]. Increasing the number of leaves along with lowering the number of tillers in the early vegetative stage could increase Nitrogen concentration in the leaves with a higher ratio of total N in the upper leaves during the later vegetative and reproductive stages, increase the carbohydrate storage capacity in stems, increase reproductive sink capacity, and extend the grain-filling capacity. This idea serves as the basis for the first and second generations of new rice ideotypes, also known as New Plant Type (NPT) Rice.

IRRI launched the Salt Tolerant Rice for Africa and South Asia (STRASA) project in 2007 in collaboration with AfricaRice, to develop rice varieties that are well adapted to abiotic stresses such as low temperatures, salinity, iron toxicity, submergence, and drought, suitable for African conditions [10]. More than a third of African countries and three South Asian countries (Bangladesh, India, and Nepal) participated in the project [33]. The STRASA project was a big success. Until the end of the project in 2019, more than 150 varieties that have tolerances to flood, drought, and salinity, including multiple tolerance improvements in mega-rice varieties, Swarna, Sambha Mahsuri, and Sahbhagi dhan, were released in South Asia and in sub-Saharan Africa. STRASA also contacted 35 million farmers and produced over 1 million tons of seeds, with 18 million hectares of covered area [33].

In terms of sustainability, Green Super Rice (GSR) initiated by the Chinese Academy of Agricultural Sciences (CAAS) in collaboration with IRRI and AfricaRice was created in 2005. The main goal of this project was to develop new rice varieties with a variety of green characteristics while retaining high and consistent production. The green character definition includes resistance to multiple biotic stresses (insects and/or diseases), external input efficiency (lower pesticide, fertilizer, and irrigation), and abiotic stress based on high grain yield and quality [34]. Over 30 Chinese institutions, universities, research centers, and private seed companies collaborated with many national agricultural research and extension system (NARES) partners from Asia and Africa to systematically introduce the GSR to 16 African and Asian countries in the breeding, adaptation testing, and capacity building stages [34]. Over a decade, 66 GSR varieties were introduced in China's five major rice-growing regions, with a total planting area of 10.87 million hectares, and 59 GSR varieties suited to unique environments were introduced in Africa, South Asia, and Southeast Asia [35].

Rice grain quality enhancement has traditionally been a primary priority in rice breeding initiatives. Eating and cooking quality has always had a major influence on market pricing and consumer approval. The rising demand for rice variants with improved grain quality and nutrition necessitates the development of a suite of rice varieties with varied combinations of attributes aimed at certain regions [36]. Adequate breeding selection strategies are essential to develop stable donor lines with better yields to successfully combine yield and grain quality [37]. Breeders are investigating the genetic regulation of both traits in IR 64, Basmati, Khao Dawk Mali, Koshihikari, and others, with the goal of transferring the traits into a mega-variety in the future. Biofortified rice with increased nutritional content will be a prevalent trend in new variety releases in the coming years. In addition to golden rice, researchers are working on rice lines that can digest starch more

slowly and rice lines with high iron and zinc content, selenium, folic acid, lysine, leucine, and other nutrients [36]. IRRI collaborated with HarvestPlus (a biofortification specialist) to release 3 zinc-rich lines in Bangladesh and more than 70 promising rice lines with higher amounts of zinc have been identified in India [38]. The improving nutrients in rice program is more reliable for implementation in developing countries because after the one-time investment to develop a rice variety, the seeds can maintain nutrient content themselves and can be shared among farmers or the community [39,40].

However, several investigations involving this characteristic did not go as expected. Genetic manipulation of the glycemic index (GI) trait results in yield losses as well as unsatisfactory cooking and texture characteristics [41]. Thus, most studies discovered a negative relationship between yield and grain Zn content, with only a few specific germplasm accessions and populations demonstrating a nonsignificant negative relationship or no relationship [42,43]. A highly significant negative correlation was found between zinc content in both brown rice and polished rice, as well as iron content in polished rice, with single plant yield [42].

*3.2. Climate-Smart Rice*

Food security is defined as a situation in which all people have physical, social, and economic access to adequate, secure, and nourishing food that always meets their nutritional requirements and preferences for a healthy and active lifestyle [44]. This has become a concern that has been on the international agenda since 1948 [45]. Climate change affects many essential aspects of food security, such as production, access, utilization, and stability. According to the FAO report, climate change would result in 71 million food-insecure people and communities around the world [46]. It is expected that in 2030 there will be 700 million small-scale agricultural producers affected and any further delay in comprehensive anticipatory global action on adaptation and mitigation would result in a missed chance to assure a livable and sustainable future for all [47].

Climate change affects temperature, water availability, floods, droughts, and is heavily reliant on rainfall patterns. The diversity, intensity, and frequency of rainfall patterns with climate change are some of the factors that cause unpredictable drought and flood conditions. The same thing happens in heat stress, where the estimated model predicts that an increase in temperature followed by drought stress will be one of the climate patterns that plants will face in the future. Catastrophic weather such as extreme droughts, temperature, flooding, and hurricanes have become more prevalent and severe because of climate change [48].

Those kinds of conditions have an impact on crop and livestock yields, hydrological balance, input supply, and other agricultural system components [45]. According to the IPCC, if current climate change effects continue unabated, Sub-Saharan areas in Africa will face 11% of agricultural output losses by 2080 [1]. While poor people are disproportionately affected by food insecurity, crop failure caused by climate variability has broader implications for poverty and food availability because food scarcity influences food prices [49].

Most future climate change models predict that warming will reduce yields for major staple crops, with tropical regions seeing greater yield losses [4]. According to a meta-analysis of climate change impacts, 70% of studies show crop output losses by 2030, with half of the studies showing 10–50% declines [50]. Achieving food security in the face of increasing food demand, competition for decreasing resources, and the environment's failing ability to buffer increasing anthropogenic consequences is commonly regarded as the most pressing challenge of our time [44].

Abiotic factors like submersion, heat, and drought greatly impede the increase in rice productivity. Although rice is very water-loving and can withstand brief periods of soaking, prolonged soaking for longer than 8–9 days has a significant effect on rice production. In the rainfed lowlands, which make up 25 Mha of the rice farming area in South and Southeast Asia, floods and submergence frequently represent a major danger to

rice production due to climate change [21]. Under the pressure of climate change, there will be a shift in pathological races or biotypes, potentially causing the resistance gene in the old varieties to be broken. The newly released rice varieties should also have different resistance to pests than the existing variety and be more durable due to the diverse genetic background of rice varieties. This strategy can compensate for crop damage and allows farmers to adjust their crop varieties over time [51].

Increases in rice output can be made sustainable by creating rice cultivars with built-in tolerance to these significant abiotic challenges. Rice, despite being a tropical plant, is extremely vulnerable to the main abiotic stresses of heat, drought, and flash floods. Climate-smart agriculture is a food system for sustainable development that aims to create potential pathways for rapid food system transformation in the context of climate change pressure. The climate-smart agriculture approach is from the grassroots upward with farmers as the main stakeholders. This is because climate change has a direct impact on farmers, and for that, farmers must understand the stages of climate change mitigation. Adoption of climate-resilient and high-yielding rice varieties can aid in this goal. Without any adaptation efforts, simulation results with temperature increases in both tropical and temperate regions of 2 °C of local warming estimated a very large loss in production, especially for the world's three main foods, namely, wheat, rice, and maize [52]. With the increase in crop-level adaptations, it is simulated that yields will increase by an average of 7–15% for the three main crops, including rice.

Artificial screening can be carried out to obtain superior genotypes that are resistant to biotic and abiotic stresses. In addition, it is also to eliminate other factors that may affect the plant, so that only the treatment or stress given will affect the plant. Many studies have been carried out on artificial screening against submergence, heat, and drought stress alone [53–63]. Rice breeding at many research centers such as IRRI has developed climate change-ready rice that can withstand unfavorable environments that occur more frequently and intensively due to climate change [64]. Climate-smart rice varieties are designed to adapt to rapidly changing climatic conditions and unfavorable areas. Drought-tolerant varieties such as Var. Sahbhagi Dhan, Var. Sahod Ulan, and Var. Sookha (Sukkha) Dhan have been developed by IRRI and are now being planted by farmers in India, the Philippines, and Nepal [64]. Under drought, the average yield advantage of drought-tolerant varieties over drought-susceptible varieties is 0.8–1.2 tonnes per hectare. Drought tolerance is being introduced into popular high-yielding rice varieties such as IR64, Swarna, and Vandna by IRRI. Swarna Sub1 in India, Samba Mahsuri in Bangladesh, IR64-Sub1 in the Philippines, and Ciherang Sub1 in Indonesia and Nepal are flood-tolerant varieties that have been released and are now being planted. Following 10–15 days of flooding, these varieties showed a yield advantage of 1–3 tonnes [64]. In practice, the adoption of climate-smart rice must be accompanied by a package regarding management practices for farming communities where the variety will be released. So, it will lead to significant increases in yield and sustainability of production in areas affected by climate change stress.

The use of farmer-accepted varieties is important in assembling CSR. These varieties usually have gone through a long selection from time to time with standard criteria that are preferred by consumers/farmers both in terms of yield and quality. However, it is known that most of the popular varieties do not tolerate environmental stress well. Thus, researchers only need to incorporate one or several additional traits into the varieties that have been released without changing the superior traits that already exist.

There have been several solutions to cope with climate change and to increase rice adaptation. Because of the environmental effects, the time required, and the low heritability of some tolerance-related traits in rice, it has proven difficult to solve using conventional breeding approaches. With recent breakthroughs in molecular biology, there are chances to address these issues through marker-assisted selection strategies for resistance or tolerance traits.

Unlike rice for irrigated ecosystems, more than 60% of the global rice production, particularly in the rainfed ecosystem, lacks high-yielding types best suited to this tough

environment. In this situation, farmers are forced to cultivate low-yielding yet well-adapted traditional cultivars. Several attempts to improve traditional varieties by introducing abiotic stress tolerance genes have failed because resistance is inherited in a complex manner and phenotypic selection is still used in selection operations.

### 3.3. Introgression Method and Gene Pyramiding in Developing Climate-Smart Rice

The effort to assemble climate-smart rice is basically one of the applications of combining several elite genes from various genotypes into a single genotype. This technique in plant breeding is called gene pyramidization. Ideally, researchers should already have information about the gene controlling the characteristic they want to combine. Identification of the morphological characteristics of candidate genotypes contributing to elite genes is needed to obtain initial information on the existence of the intended gene. The pyramiding strategy entails crossing with chosen parental lines, followed by the selection of hybrids and their progeny for the desired trait [65].

The most efficient method of developing multi-tolerant rice consisting of multiple stress tolerances is to combine genotype selection involving gene introgression in popular varieties with phenotype-based selection [2,5,6,66–70]. With this method, researchers usually insert one or two specific genes into a genotype that has stable characteristics. Attempts to combine genes in conventional breeding are carried out through artificial hybridization from selected gene source parents, followed by selection actions based on phenotypic appearance. The process is not only limited by a long time requirement but also by the possibility of other unwanted traits besides the desired genes, still appearing even after multiple generations [71]. Pedigree, backcrossing, or recurrent selection are commonly used to transfer inherited features and resistance genes from donor parents onto recipient lines.

Pedigree breeding is commonly used to improve the characteristics of self-pollinated species, in which superior individual plants are selected from F2 and the next generations as their progenies are grown, and the parent–progeny relationship (pedigree) data are kept at each stage of selection [65]. Meanwhile, multiple cycles of selection and breeding are used in recurrent selection with the purpose of gradually genetically improving a few essential features in a single species.

The hybrid and its offspring are regularly backcrossed to one of the parents (usually the popular variety) in the backcross method. Therefore, unless the characteristic is fixed, the backcross progeny's genotype grows progressively like that of the recurrent parent. This method is highly efficient for transferring a single or few targeted genes from separate parents into a single genotype in a shorter time (two to three generations). The traits of interest are determined through the selection process in the backcrossing procedure.

Many researchers have employed backcrossing to carry out resistance gene pyramiding. Gene pyramiding enhances the possibility of crops developing long-term tolerance to biotic and abiotic stressors [72]. Combining various resistances is thought to increase resistance to a wide range of pressures in one type.

Figure 2 explains that there are two stages in the gene pyramiding scheme culminating in multiple genes at once: pedigree and gene fixation [73]. At the pedigree stage, these desirable genes are selected from various genotypes in the founding parents. The selected founding parent is then crossed through a single cross with another parent. The offspring of one pair of crosses (called the intermediate genotype) is then crossed with another intermediate genotype. A progeny with all of the target genes from both parents is chosen as an intermediate genotype. Thus, the intermediate genotype variation has resistance. Finally, all target genes can be collected in a single genotype called the root genotype.

The second stage is known as the fixation step, and it is used to fix up the target genes in the homozygous condition to obtain the optimum genotype from a single genotype. Traditionally, the fixation of crossed genes is conducted by crossing themselves until they reach the ideal genotype, but with crossing over, it is possible that there are broken genes so that more generations are needed. This phase can be accomplished effectively by

crossing the root genotype with one of the non-donor parents, also known as backcross gene pyramiding.

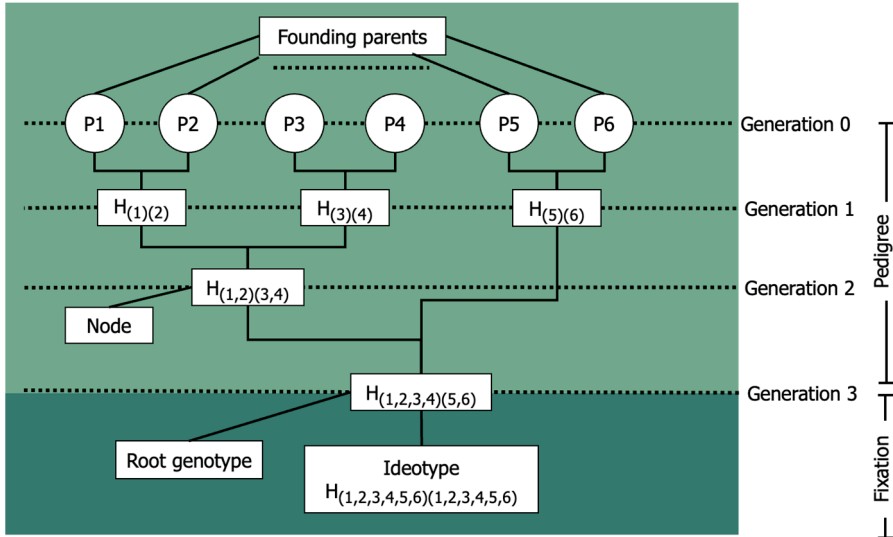

**Figure 2.** Scheme of gene pyramiding for accumulating six target genes [73]. "Reproduced with permission from Servin et al., 2004, Genetics; published by Oxford University Press, 2004".

This method is the most efficient method in assembling multi-tolerant rice to combine genotype selection involving desirable genes in popular varieties with phenotype-based selection. The genotype that receives the desirable gene is usually called the recurrent genotype. It is usually a commercial cultivar that has been accepted and has been widely used by farmers. The genotype with the introgression gene is called the donor parent. The donor parent can come from any genotype that is known to contain any gene of interest (both from the same species or closely related species). With the development of technology, donor gene sources can be obtained from unrelated species using a gene transformation approach.

Several factors contribute to the success of gene pyramiding, including information about the genes to be integrated (the number of genes to be transferred and their distance from the flanking markers), the pattern of inheritance of these genes in each generation, and other genetic parameters.

Inheritance studies of submergence- and flood-tolerant characteristics in rice show that the genes are controlled by one dominant gene [74]. Meanwhile, tolerance to drought stress is controlled by many genes with low-to-moderate heritability [75]. Other studies have shown that tolerance to heat stress at the flowering stage of rice is controlled by recessive genes [68,76].

Figure 3 describes that there are three schemes of backcrossing that can be conducted to pyramidize genes: stepwise transfer, simultaneous transfer, and combination of simultaneous and stepwise transfer [77]. In the first scheme, the recurrent parent (RP) is crossed with one of the donor parents (DP1) to produce the F1 hybrid. The backcrossing is conducted up to the third backcross generation (BC3) to produce the improved recurrent parent (IRP). The next step is crossing the improved recurrent parent with a different donor parent (DP2), resulting in the pyramiding/stacking of numerous genes. This method takes more time, but the pyramiding result can be incredibly exact because it only involves one gene at a time.

In the simultaneous transfer scheme, F1 hybrid plants are created by separately crossing the one recurrent parent (RP) with two different donor parents (DP1 and DP2). Intercrossing the two F1 generations results in improved F1 (IF1) hybrids. Following that, backcrossing between the recurrent parents (RP) and the improved F1 will be undertaken to produce improved recurrent parents (IRP). This type of pyramiding occurs during the

pedigree stage when the donor parents are not the same. This strategy, however, is less likely to be used because the pyramided gene may be lost in the process.

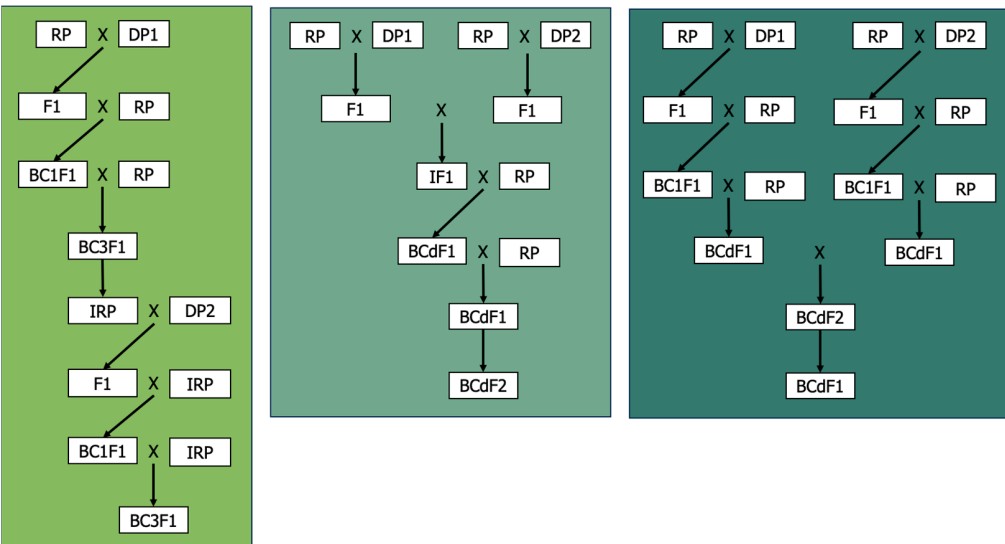

**Figure 3.** Scheme of backcrossing for gene pyramiding. Left–right: stepwise transfer, simultaneous transfer, combination of simultaneous and stepwise transfer [77].

The last scheme is the most accepted strategy because it not only saves time but also ensures complete gene fixation. It combines the previous two by crossing one recurrent parent (RP) with several donor parents (DP1 and DP2) and then backcrossing each F1 with RP up to the BC3 generation separately. The pyramided lines are created by intercrossing each backcross population. This is the most accepted strategy because it not only saves time but also ensures complete gene fixation.

However, the implementation of artificial hybridization requires special skills, information about the flower anthesis for each genotype, and researchers must ensure that the characteristics to be transferred are not associated with unfavorable characteristics. This makes conventional hybridization techniques more complicated. The introgression of characteristics controlled by recessive genes will require one more season for progeny testing and for selecting progenies that do contain the gene of interest [71]. For this reason, the approach is carried out through marker-assisted selection (MAS) and genetic transformation approaches.

Marker-assisted selection (MAS) is one indirect selection method using a linked marker to the target gene for the tagging of some important agronomic traits such as for resistance and tolerance to biotic or abiotic stresses. This strategy also enables the breeder to select desirable or undesired DNA/genes from in between backcrosses, as early as in plant growth, saving time by reducing breeding generations. QTL and development marker genetic mapping have increased the use of molecular techniques in rice breeding. QTLs influence the phenotypic variation of a complex trait, frequently through genetic interactions with each other and the environment [78]. Quantitative trait loci are genes that may be strong candidates for selection; nevertheless, it is difficult to find inheritance at QTLs alone.

In phenotypic selection, plant genotypes are selected on the basis of their phenotype for one or more characteristics. The selected individuals are self-tested and tested crossed with a tester simultaneously. When combined with MAS, plant genotypes are selected based on the specific molecular markers attached to the genes or QTLs and the intercrosses among the selected individuals are also made in the same crop season for one cycle of selection. Thus, this method has been shown to increase time, effort, and accuracy efficiency in various rice breeding activities [2–6,79–81].

The breeding of environmentally stress-tolerant rice has been extensively studied [67,69,82–84]. However, due to the difficulties of identifying matched donors

with a greater tolerance level as well as the special characteristics of their surroundings, the success percentage falls significantly short of expectations. Several QTLs for tolerance to environmental problems have been successfully identified and introduced into superior cultivars using a marker-assisted breeding technique (Table 1).

**Table 1.** Several rice QTLs for tolerance to environmental problems have been successfully identified using a marker-assisted selection technique.

| QTL/Gene | Target Trait | Chromosome | Population Observed | Marker Used | Reference |
|---|---|---|---|---|---|
| *Sub1* | Submergence tolerance | 1,2,9,12 | IR72/Madabaru | SSR | [66] |
| *qGL3-1* | Grain size | 3 | Xihui 18/Hunan-3 (japonica) | SSR | [85] |
| *qGL3-2* | Grain size | 3 | Xihui 18/Hunan-3 (japonica) | SSR | [85] |
| *qGL7* | Grain size | 7 | Xihui 18/Hunan-3 (japonica) | SSR | [85] |
| *qGW5* | Grain width | 5 | Asominati/IR24 | SSR | [86] |
| *qTGW12a* | Grain weight | 12 | indica | SNP | [87] |
| *qHI-8* | Harvest index | 8 | Yuexiangzhan (indica)/Shengbasimiao (indica) | SSR | [88] |
| *qFe2.1* | Fe and Zn for yield and quality | 2 | Swarna/IRGC81832(*O. nivara*), Swarna/IRGC81848 (*O. nivara*) | SSR | [89] |
| *qFe3.1* | Fe and Zn for yield and quality | 3 | Swarna/IRGC81832(*O. nivara*), Swarna/IRGC81848 (*O. nivara*) | SSR | [89] |
| *qFe8.2* | Fe and Zn for yield and quality | 8 | Swarna/IRGC81832(*O. nivara*), Swarna/IRGC81848 (*O. nivara*) | SSR | [89] |
| *qZn12.1* | Fe and Zn for yield and quality | 12 | Swarna/IRGC81832(*O. nivara*), Swarna/IRGC81848 (*O. nivara*) | SSR | [89] |
| *qDTY11* | Grain yield under drought | 11 | Nagina 22/MTU1010, Nagina 22/Swarna, Nagina 22/IR64 | SSR | [90] |
| *qTWU3-1* | Total water uptake under drought | 3 | IR55419-IR55419-04/Super Basmati | SSR | [91] |
| *qGYD1.1* | Grain yield under drought | 1 | Cocodrie/Vandana (japonica) | SSR and SNP | [92] |
| *qGYD1.2* | Grain yield under drought | 1 | Cocodrie/Vandana (japonica) | SSR and SNP | [92] |
| *qGYD1.3* | Grain yield under drought | 1 | Cocodrie/Vandana (japonica) | SSR and SNP | [92] |
| *qGYD5.1* | Grain yield under drought | 5 | Cocodrie/Vandana (japonica) | SSR and SNP | [92] |
| *qGYD8.1* | Grain yield under drought | 8 | Cocodrie/Vandana (japonica) | SSR and SNP | [92] |
| *qGYD9.1* | Grain yield under drought | 9 | Cocodrie/Vandana (japonica) | SSR and SNP | [92] |
| *DRO1* | Deeper rooting under drought stress | 9 | IR64/Kinandang Patong | SSR and SNP | [93] |
| *DRO3* | Deeper rooting under drought stress | 7 | IR64/Kinandang Patong | SSR and SNP | [94] |
| *DRO2* | Deeper rooting under drought stress | 4 | Kinandang Patong (KP)/japonica | SSR and SNP | [95] |
| *qtl12.1* | Plant water uptake under drought stress | 12 | Vandana/Way Rarem | SSR | [96] |

**Table 1.** *Cont.*

| QTL/Gene | Target Trait | Chromosome | Population Observed | Marker Used | Reference |
|---|---|---|---|---|---|
| *qDTY2.1* | Grain yield under drought stress | 2 | Apo/Swarna, Aday sel/IR64, Way Rarem/Vandana | SSR | [97] |
| *qDTY2.2* | Grain yield under drought stress | 2 | Apo/Swarna, Aday sel/IR64, Way Rarem/Vandana | SSR | [97] |
| *qDTY9.1* | Grain yield under drought stress | 9 | Apo/Swarna, Aday sel/IR64, Way Rarem/Vandana | SSR | [97] |
| *qDTY12.1* | Grain yield under drought stress | 12 | Apo/Swarna, Aday sel/IR64, Way Rarem/Vandana | SSR | [97] |
| *qDTY2.2* | Grain yield under drought stress | 2 | Aday Sel/IR64 | SSR | [98] |
| *qDTY4.1* | Grain yield under drought stress | 4 | Aday Sel/IR65 | SSR | [98] |
| *qDTY9.1* | Grain yield under drought stress | 9 | Aday Sel/IR66 | SSR | [98] |
| *qDTY10.1* | Grain yield under drought stress | 10 | Aday Sel/IR67 | SSR | [98] |
| *qDTY1.1* | Grain yield under drought stress | 1 | N22/Swarna, N22/IR64, and N22/MTU1010 | SSR | [90] |
| *qDTY1.1* | Grain yield under drought stress | | Dhagaddeshi/Swarna, Dhagaddeshi/IR64 | SSR | [99] |
| *qST1.1* | Salt tolerance | 1 | Dianjingyou1/Sea rice 86 (indica) | SSR | [100] |
| *qSH1.3* | Seed height under salt stress | 13 | Juicaiqing (japonica)/IR26 | SSR | [101] |
| *qSH12.1* | Seed height under salt stress | 12 | Juicaiqing (japonica)/IR26 | SSR | [101] |
| *qSH12.2* | Seed height under salt stress | 12 | Juicaiqing (japonica)/IR26 | SSR | [101] |
| *qDSW12.1* | Dry shoot weight under salt stress | 12 | Juicaiqing (japonica)/IR26 | SSR | [101] |
| *qDRW11* | Dry root weight under salt stress | 11 | Juicaiqing (japonica)/IR26 | SSR | [101] |
| *qSIS1* | Salt injury score | 1 | IR29/Pokkali | SNP | [102] |
| *qSIS4* | Salt injury score | 4 | IR29/Pokkali | SNP | [102] |
| *qSIS12* | Salt injury score | 12 | IR29/Pokkali | SNP | [102] |
| *qRSL7* | Relative shoot length under salt stress | 7 | IR36/Weiguo | SNP | [103] |
| *SKC1* | Maintain K+ homeostasis under salt stress | 1 | Nona Bokra (indica)/Koshihikari (japonica) | SSR | [104] |
| *qST-3.1* | Salt tolerance | 3 | RPY geng/Luohui 9 | RNA-seq | [105] |
| *qST-5.1* | Salt tolerance | 5 | | RNA-seq | [105] |
| *qST-6.1* | Salt tolerance | 6 | | RNA-seq | [105] |
| *qST-6.2* | Salt tolerance | 6 | | RNA-seq | [105] |
| *qGY11* | Grain yield under salt stress | 11 | Nerica-L-19/Hasawi, Sahel 108-/Hasawi, BG902/Hasawi | SSR | [106] |

**Table 1.** *Cont.*

| QTL/Gene | Target Trait | Chromosome | Population Observed | Marker Used | Reference |
|---|---|---|---|---|---|
| *qTN11* | Tillering number under salt stress | 11 | Nerica-L-19/Hasawi, Sahel 108-/Hasawi, BG902/Hasawi | SSR | [106] |
| *qAT11* | Alkalinity | 11 | Xiaobaijingzi (japonica) | SNP | [107] |
| *qCTS12* | Cold tolerance | 12 | M202 (japonica)/IR50 | SSR | [108] |
| *qCTS12a* | Cold-induced wilting tolerance (CIWT) | 12 | M202 (japonica)/IR50 | SSR | [108] |
| *qCGR8* | Gult rate after cold stress | 8 | Changhui 891/02428//02428 | SNP | [109] |
| *qGRR11* | Seed germination recovery rate after cold stress | 11 | Changhui 891/02428//02428 | SNP | [109] |
| *qNGR1* | Germination under normal condition | 1 | Changhui 891/02428//02428 | SNP | [109] |
| *qNGR4* | Germination under normal condition | 4 | Changhui 891/02428//02428 | SNP | [109] |
| *qGRR11* | Germination recovery rate after cold stress | 11 | Changhui 891/02428//02428 | SNP | [109] |
| *qGRR8* | Germination recovery rate after cold stress | 8 | Changhui 891/02428//02428 | SNP | [109] |
| *qCTSL8-1* | Cold tolerance at seedling stage with leaf discoloration | 8 | BR1/Hbj.BVI (indica) | SSR | [110] |
| *qCTSS8-1* | Cold tolerance at seedling stage with survival rate | 8 | BR1/Hbj.BVI (indica) | SSR | [110] |
| *qCTSL12-1* | Cold tolerance at seedling stage with leaf discoloration | 12 | BR1/Hbj.BVI (indica) | SSR | [110] |
| *qCTSS-12-1* | Cold tolerance at seedling stage with survival rate | 12 | BR1/Hbj.BVI (indica) | SSR | [110] |
| *qCTBB9* | Cold tolerance at the bud bursting | 9 | Dongnong 430 (DN430) and Dongfu104 (DF104) (japonica) | SNP | [111] |
| *qLTG-3* | Germination rate at low temp | 3 | Heigu and Ha 9366 | SSR | [112] |
| *qLTG-12* | Germination rate at low temp | 12 | Heigu and Ha 9366 | SSR | [112] |
| *(qLTG(I)1* | Germination rate at low temp | 1 | Weed Tolerant Rice-1 (WTR-1) and Haoannong (HNG) | SNP | [113] |
| *qLTGS(I)1–2* | Stress index at low temp | 1 | Weed Tolerant Rice-1 (WTR-1) and Haoannong (HNG) | SNP | [113] |
| *qLTG(I)5* | Germination rate at low temp | 5 | Weed Tolerant Rice-1 (WTR-1) and Haoannong (HNG) | SNP | [113] |
| *qLTGS(I)5* | Stress index at low temp | 5 | Weed Tolerant Rice-1 (WTR-1) and Haoannong (HNG) | SNP | [113] |
| *qLTG(I)7* | Germination rate at low temp | 7 | Weed Tolerant Rice-1 (WTR-1) and Haoannong (HNG) | SNP | [113] |
| *qHTT8* | Heat tolerance T-pool type | 8 | Huanghuazhan (indica)/9311 (indica) | SNP and InDel | [114] |

**Table 1.** *Cont.*

| QTL/Gene | Target Trait | Chromosome | Population Observed | Marker Used | Reference |
|---|---|---|---|---|---|
| *qHTSF4.1* | Spikelet fertility under heat stress | 4 | IR64/N22 | SNP | [115] |
| *qHTSF1.2* | Spikelet fertility under heat stress | | IR64/Giza178 | SNP | [76] |
| *qHTSF2.1* | Spikelet fertility under heat stress | 1 | IR64/Giza178 | SNP | [76] |
| *qHTSF3.1* | Spikelet fertility under heat stress | 4 | IR64/Giza178 | SNP | [76] |
| *qHTSF4.1* | Spikelet fertility under heat stress | 6 | IR64/Giza178 | SNP | [76] |
| *qHTSF6.1* | Spikelet fertility under heat stress | 11 | Milyang23/Giza17 | SNP | [76] |
| *qHTSF11.2* | Spikelet fertility under heat stress | 11 | Milyang23/Giza17 | SNP | [76] |
| *qHTSF4.1* | Spikelet fertility under heat stress | 4 | Takanari/IR64 | SNP | [116] |
| *qSSR6-1* | Seed-setting rate under heat stress | 6 | N22/9311 | SSR | [117] |
| *qSSR7-1* | Seed-setting rate under heat stress | 7 | N22/9312 | SSR | [117] |
| *qSSR8-1* | Seed-setting rate under heat stress | 8 | N22/9313 | SSR | [117] |
| *qSSR9-1* | Seed-setting rate under heat stress | 9 | N22/9314 | SSR | [117] |
| *qSSR11-1* | Seed-setting rate under heat stress | 11 | N22/9315 | SSR | [117] |
| *qSTIPSS9.1* | Spikelet sterility under heat stress | 1 | Nagina22/IR64 | SNP | [118] |
| *qSTIY5.1/qSSIY5.2* | Percent spikelet sterility and yield per plant under heat stress | 5 | Nagina22/IR65 | SNP | [118] |
| *qHTH5* | Heat tolerance at the heading stage | 5 | *O. rufipogon* Griff | SNP | [119] |
| *qEMF3* | Early-morning flowering under heat stress | 3 | EMF20/Nanjing 11 | SSR | [120] |
| *ZFP gene* | Heat tolerance gene at seedling stage | 9 | Guang-Lu-Ai No 4/HT13 | SSR and InDel | [121] |
| *qSE3* | Seed germination and seedling establishment under salinity stress | | Jiucaiqing (japonica), IR26 and 62 CSSLs | SNP | [122] |
| *Sub 1* | Submergence tolerance | | IR40931-26, IR40931-2 (indica)/PI543851 (japonica) | RAPD, AFLP, and RFLP | [123] |
| *Sub1A gene* | Flash flood tolerance | 9 | DX18-121(indica)/M-202 (japonica) | | [124] |
| *SD1-DW* | Deepwater rice-specific variant of SEMIDWARF1 | | | | [125] |

**Table 1.** *Cont.*

| QTL/Gene | Target Trait | Chromosome | Population Observed | Marker Used | Reference |
|----------|--------------|------------|---------------------|-------------|-----------|
| *LGF1/OsHSD1* | Leaf gas films under flood | | Dripping wet leaf 7,drp7 (Kinmaze mutant) | CAPS marker | [126] |
| *OsTPP7* | trehalose-6-phosphate phosphatase gene | | | | [127] |
| *qAG-1-2* | Flood tolerance during germination | 1 | IR64/Khao Hlan On | SSR and InDel | [128] |
| *qAG-3-1* | Flood tolerance during germination | 3 | IR64/Khao Hlan On | SSR and InDel | [128] |
| *qAG-7-2* | Flood tolerance during germination | 7 | IR64/Khao Hlan On | SSR and InDel | [128] |
| *qAG-9-1* | Flood tolerance during germination | 9 | IR64/Khao Hlan On | SSR and InDel | [128] |
| *qAG-9-2* | Flood tolerance during germination | 9 | IR64/Khao Hlan On | SSR and InDel | [128] |
| *qTIL12* | Total internode elongation length under flood stress | 12 | T65/C9285, T65/W0120W0120 (*O. rufipogon*)/C9285 (indica) | SSR and CAPs | [129] |
| *qNEI12* | Number of elongated internodes under flood stress | 12 | T65/C9285, T65/W0120W0120 (*O. rufipogon*)/C9285 (indica) | SSR and CAPs | [129] |
| *qLEI12* | Position of the lowest elongated internode under flood stress | 12 | T65/C9285, T65/W0120W0120 (*O. rufipogon*)/C9285 (indica) | SSR and CAPs | [129] |
| *Sub 1* | Submergence tolerance | 9 | RILs IR74 (indica)+FR13A (indica) | AFLP | [130] |
| *SK1 and SK2 genes* | Snorkel genes under deepwater condition | | W0120, C9285, Bhadua, W0106, T65 and *O. glumaepatula* | | [131] |
| *Sub1* | Submergence tolerance | 1,2,9,12, | IR72/Madabaru (IRGC 15333) | SSR | [66] |
| *OsTPP1, LOC_Os02g44230* | trehalose-6-phosphate phosphatase gene under O2-deficient conditions on anaerobic germination | 1 | H335/HA-1 | SNP | [132] |
| *qAG-2* | Anaerobic germination | 2 | IR42/Ma-Zhan Red | SSR | [133] |
| *qAG-5* | Anaerobic germination | 5 | IR42/Ma-Zhan Red | SSR | [133] |
| *qAG-6* | Anaerobic germination | 6 | IR42/Ma-Zhan Red | SSR | [133] |
| *qAG-7-1* | Anaerobic germination | 7 | IR42/Ma-Zhan Red | SSR | [133] |
| *qAG7.1* | Anaerobic germination | 7 | IR64/Kharsu 80A | SNP | [134] |
| *qAG7.2* | Anaerobic germination | 7 | IR64/Kharsu 80A | SNP | [134] |
| *qAG7.3* | Anaerobic germination | 7 | IR64/Kharsu 80A | SNP | [134] |
| *qAG3* | Anaerobic germination | 3 | IR64/Kharsu 80A | SNP | [134] |
| *qAG2.1* | Anaerobic germination | 2 | IR64/Nanhi | SNP | [135] |
| *qAG11* | Anaerobic germination | 11 | IR64/Nanhi | SNP | [135] |
| *qAG7* | Anaerobic germination | 7 | IR64/Nanhi | SNP | [135] |

In the last twenty years, major genes and QTLs related to tolerance to rice environmental stresses along with markers that can be in the selection process have been

identified [54,66,67,82,120,136–139]. The *Sub1* gene has been widely recognized as a QTL associated with submergence and flood tolerance [83,140–143]. Several rice QTLs related to increased production (*qDTY*) under drought stress have also been identified on chromosomes 1, 2, 3, 4, 6, 9, 10, and 12 [97–99,144,145]. Drought-related QTLs (*qDRL*) were identified on chromosomes 8 and 9 [82,97]. Spikelet sterility caused by heat exposure has been identified by the presence of the *qHTSF* QTL on chromosomes 1, 3, 4, 6, and 11 [115].

Quantitative trait locus (QTL) hotspots are specific regions within a genome where multiple QTLs associated with different desirable traits are clustered or co-located. In other words, these are regions of the genome where multiple genes or genetic markers that influence various quantitative traits of interest are found close together. The Swarna*3/Morobekerjaan-derived population was studied in order to identify six genomic areas for early vigor and related traits, two of which were QTL hotspots (QTL hotspot A and QTL hotspot B), which harbored early vigor and related traits under a dry direct-seeded system [146]. Recently, the presence of a QTL hotspot that was strongly associated with drought tolerance was detected on rice chromosome 8 in the RIL population derived from rice IR64 × Hawara Bunar using differential gene expression meta-analysis and the qRT-PCR technique [147].

QTL hotspots are of great importance in the context of gene pyramiding, which is the process of stacking multiple desirable traits into a single crop variety. QTL hotspots make it easier to combine multiple desirable traits in a single breeding program. Because several QTLs are concentrated in one genomic region, breeders can target that region to introgress multiple traits simultaneously, reducing the number of generations and resources needed for pyramiding. QTL hotspots can help minimize linkage drag because the multiple traits of interest are close together, allowing breeders to select for the specific QTLs they want while avoiding those associated with undesired traits. By leveraging QTL hotspots, breeders can develop improved crop varieties more rapidly. This is particularly valuable when breeding for complex traits, such as disease resistance, where multiple genes may be involved. Hotspots allow breeders to identify a genomic region associated with resistance to multiple diseases, thereby streamlining the breeding process. QTL hotspots enable more precise breeding strategies. Breeders can use marker-assisted selection (MAS) or gene editing techniques to specifically target the QTLs within the hotspot, ensuring that the desired traits are incorporated into the final variety without unnecessary genetic baggage. Stacking multiple QTLs from a hotspot can lead to synergistic effects, resulting in crop varieties with enhanced performance. Combining QTLs for high yield and disease resistance from a hotspot, for example, can yield varieties with both traits, providing a more productive and resilient crop.

Several studies by other researchers carried out genetic analysis for heat stress during flowering time, the response on formation of fertile spikelets, and quality characteristics concerning the chalkiness of grain [148]. Identification of QTLs is useful for stress tolerance in the early and vegetative growth phases of rice, as well as candidate genes using the genotyping-by-sequencing (GBS) approach to compose SNP markers that will be used for genotype selection and map preparation [137]. It recognizes the genetic mechanism of tolerance to heat stress during rice anthesis in relation to pollen fertility [116] and the genes that were expressed under heat conditions [149]. Several QTLs related to the nature and response of rice to heat stress, including the *qHTSF* associated with spikelet sterility, have been confirmed to be located on chromosomes 1, 4, 6, and 11 from the cultivars IR64/N22, Giza 178, and MY23, while QTL *qHTH* was identified on chromosome 5 of *O. rufipogon* [68,115,116].

Floods, one of rice's other major environmental stresses, have enabled researchers to identify the existence of *qSUB1* which is confirmed to be located on chromosome 9 and is associated with reduced elongation growth and carbohydrate consumption during submergence. The QTL *qTIL* identified on chromosome 12 and chromosome 1 is known to affect the rapid elongation of internodes up to 20–25 cm per day and up to a few meters to escape submergence [129].

Genetic markers are useful selection tools for choosing a superior genotype, as well as target genes with effects that are difficult to see phenotypically (the recessive genes, multiple tolerance/resistance from gene pyramids, and so on). In the backcrossing method, these markers can be used to select the offspring that harbored the donor allele and still maintain the DNA composition of the recipient parent. MAS has been extensively used mainly for marker-assisted backcross selection (MABS), including the pyramiding of useful genes/QTLs, and for marker-assisted recurrent selection (MARS). Several successfully pyramided genes and their traits for biotic, abiotic stresses, and quality improvements in rice are shown in Table 2.

Several successful attempts on the use of MABS in inserting QTLs related to environmental stress have been widely published. Researchers utilized MABS in transferring several QTLs related to drought-related stress into various mega-varieties, such as IR64, Malaysia's elite drought-resistant rice cultivar MR219, and TDK1 for high yield under drought [53,67,83].

Table 2. Successfully pyramided genes for biotic, abiotic stresses, and quality improvements in rice.

| Recipient Genotype | QTL/Gene Transferred | Traits Transfer/Pyramiding | Donor Genotype | Method of Transfer | Marker Used | Country | Ref. |
|---|---|---|---|---|---|---|---|
| Tainung82 (japonica) | *xa4*, *xa5*, *xa7*, *xa13*, *xa21* | Bacterial blight resistance | IRBB66 | MABB | Xa4F/4R, RM604F/604R, Xa7F/7-1R/7-2R, Xa13F/13R, and Xa21F/21R | Taiwan | [150] |
| Jin 23B | *Bph3*, *Bph14*, *Bph15*, *Bph18*, *Bph20*, *Bph21* | Brown planthopper resistance | PTB33, IR65482–7–216-1-2, IR71033–121-15, B5 | MARS | SSR: nRM58, RM19324, RM3331, RM28427, RM28561, RM16553, HJ34, RR28561, B212 | China | [151] |
| Jalmagna | *xa5*, *xa13*, *xa21* | Bacterial blight resistance | CRMAS 2232–85 (Swarna and IRBB 60 cross derivative) | MABB | SSR: Xa5S (Multiplex), Xa5SR/R (Multiplex), RG136, pTA248 | India | [152] |
| CO39 | *Pi1*, *Piz-5a*, *Pita* | Blast resistance | C101LAC, C101A51, C101PKT | MAS for pyramiding | RFLP: RZ536, RZ64, RZ612, RG456, RG869, RZ397 | Philippines | [153] |
| Jin 23B | *Pi1*, *Pi2*, *D12* | Blast resistance | BL6, Wuyujing 2 | MABB | SSR: RM144, RM224, PI2-4, HC28, RM277, M309 | China | [154] |
| Pusa RH10 | *xa13*, *xa21* | Bacterial blight resistance | Pusa1460 (Basmati Rice) | MABB | STMS: RG136, pTA248 | India | [155] |
| Ningjing3 (Japonica) and Indica 93-11 | *Bph3*, *Bph27 (t)* | BPH resistance | Balamawe | MABB | RH078, RH7, Q31, Q52, Q58, RM471 | China | [156] |
| Swarna-Sub1 | *Pi1*, *Pi2*, *Pi54* | Blast resistance | Swarna-LT, Swarna-A51 | MABB | SSR: RM224, RM527, RM206, PI54 MAS | India | [157] |
| C101A51, Tetep | *Piz5*, *Pi54* | Blast resistance | PRR78 | Phenotypic selection | AP5930, RM206, RM6100 | India | [158] |
| Tapaswini | *xa5*, *xa13*, *xa21*, *xa4* | Bacterial blight resistance | IRBB 60 | MABB | SSR | Philippines | [159] |
| Lalat | *xa5*, *xa13*, *xa21*, *xa4* | Bacterial blight resistance | IRBB 60 | MABB | SSR | Philippines | [160] |
| Mangeumbyeo (japonica) | *xa4*, *xa5*, *xa21* | Bacterial blight resistance | IRBB57 | MABB | SSR | South Korea | [161] |
| Weed Tolerance Rice 1 (WTR1) | qSES2, GPDQ3, qSES4, and qChlo4 | Yield, submergence, drought tolerance | Khazar (aromatic type) and BG300 | MABB | SNP | Philippines | [162] |
| MRQ74 | *qDTY2.2*, *qDTY3.1*, *qDTY12.1* | Drought tolerance | IR77298-14-1-2-10, IR81896-B-B-195, IR84984-83-15-18-B | MABB | SSR: RM12460, RM511, RM520 E16 | Malaysia | [53] |
| ADT43 | *Pi54*, *Pi33*, *Pi1*, *Pi2* | Blast resistance | CT 13432-3R | MABB | SSR: RM206, RM72, RM527, RM1233 | India | [163] |

**Table 2.** *Cont.*

| Recipient Genotype | QTL/Gene Transferred | Traits Transfer/Pyramiding | Donor Genotype | Method of Transfer | Marker Used | Country | Ref. |
|---|---|---|---|---|---|---|---|
| R175 (indica) | *Pi2*, *Pi46*, *Pita* | Blast resistance | H4, Huazhan | MABB | SSR: RM224, Indel: Ind306, Pita-based Marker: Pita-Ext Pita-Int | China | [4] |
| Suisei | *qCTF7*, *qCTF8*, *qCTF12* | Cold tolerance at the fertilization stage | Eikei88223 | MABB | SSR | Japan | [164] |
| Hua-jing-xian 74 (indica) | *qCTBB-5*, *qCTBB-6*, *qCTS-6*, *qCTS12* | Cold tolerance at bud bursting and seedling stage | Nan-yangzhan (japonica) | MABB | SSR: RM170, RM589, RM17, RM31 | China | [165] |
| Swarna+drought (indica) | *xa4*, *xa5*, *xa13*, and *xa21* (BLB), *Pi9* (Blast), *Bph3* and *Bph17* (BPH), *Gm4* and *Gm8* (gall midge) | Blast+BLB+BPH+GM+DT | IRBB60, IRBL9, Rathu Heenati, Abhaya and Aganni | MAFB (FGS) | SSR | Philippines | [2] |
| White Ponni | *qDTY1.1*, *qDTY2.1*, *Saltol*, *Sub1* | Drought, salinity, and submergence tolerance | Apo (*DTY*), Pokkali FL478 (saltol), FR13A (*Sub1*) | MABB (FGS+BGS) | SSR | India | [143] |
| IR9784-226-335-1-5-1-1 (UKM5) | *qDTY3.1*, *qDTY12.1*, *Sub2* | Drought and submergence tolerance | IR64-Sub1 | MABB (FGS) | SSR | Malaysia | [166] |
| IR74 | | Submergence tolerance | FR13A | AFLP | AFLP | Philippines | [167] |
| Putra-1 (Pi9, Pi2 and Piz blast R-genes) | *xa4*, *xa5*, *xa13*, *xa21* | Blight resistance | IRBB60 | MABB (FGS) | SSR | Malaysia | [168] |
| Tella-hamsa | *xa21*, *xa13*, *Pi54*, *Pi1* | Blight and blast resistance | Improved Samba Mahsuri, Swarnamukhi | MABB | SSR: pTA248, Xa13 Prom, r Pi54MAS, RM 224 | India | [169] |
| JGL1798 | *xa13*, *xa21*, *Pi54* | Blight and blast resistance | Improved Samba Mahsuri, NLR145 | MABB | Xa13-prom, pTA248, Pi54-MAS | India | [170] |
| ASD 16 and ADT 43 | *xa5*, *xa13*, *Xa21 Pi54*, *qSBR7-1*, *qSBR11-1*, *qSBR11-2* | Blight, blast and sheath blight resistance | IRBB60, Tetep | MABB | pTA248, Xa13-prom, Xa5, Pi54-MAS RM224, RM336, RM209 | India | [171] |
| RPHR-1005 | *Xa21*, *Gm4*, *Gm8*, *Rf3*, *Rf4* | Blight and gall midge resistance | Improved Samba Mahsuri | MABB | pTA248, LRR-del, PRP, DRRM-Rf3–10, DRCG-RF4–14 | India | [172] |

**Table 2.** *Cont.*

| Recipient Genotype | QTL/Gene Transferred | Traits Transfer/Pyramiding | Donor Genotype | Method of Transfer | Marker Used | Country | Ref. |
|---|---|---|---|---|---|---|---|
| Improved Lalat (*xa4*, *xa21*, *xa13*, and *xa5*) | *Pi2* and *Pi9*; *Gm1* and *Gm4*; *Sub1*; *Saltol* | Blast resistance, all midge resistance, submergence resistance, salinity resistance | WHD-1S-75-1-127 (*O. minuta* derivative), Kavya, Abhaya, FR13A, FL478 | MABB | STS and SSR | India | [5] |
| Pyramided lines | *Sub1*, *badh2*, *qBl1*, *qBl11* | Flash flooding tolerance, fragrance quality, blast resistance | IR85264, TDK303, RGD07529 | MAS for pyramiding | R10783indel, RM212-RM319, RM224-RM144, Aromarker | Lao PDR | [173] |
| Jumam (japonica) (cold tolerance) | *xa4* | Blight resistance | IR72 (indica) | MABB | SSR: RM1233, RM3577, RM4112, RM5766, RM224 | Korea | [174] |
| Pink3 | *Sub1A-C*, *SSIIa*, *Xa5*, *Xa21*, *TPS*, *qBph3*, *qBL1*, *qBL11* | Submergence tolerance, blight resistance, blast resistance, BPH resistance | CholSub1, Xa497, RBPiQ, Bph162 | MABB | SNP and SSR markers | Thailand | [6] |
| Wuyujing 3 | Stv-bi, Wx-mq | Stripe disease resistance and low-amylose content | Kanto 194 (japonica) | MABB | ST-10, Wx-mq-OF, Wx-mq-IR | China | [175] |
| Jumam (Japonica) | *Bph18*, *qSTV11SG*, *Pib*, *Pi*, *xa3*, *xa40* | BPH resistance, stripe virus resistance, blast resistance, blight resistance | IR65482–7–216-1-2 (indica) | MABB | 7312.T4A+HinfI, ID55.WA3, RM1233, 10571.T7+HinfI, NSB, K6415, Indel7 | Korea | [176] |
| Pusa Basmati | *xa13*, *xa21*, *Pi54*, *Piz-5* | Blight resistance; blast resistance | IRBB55 (an isogenic line of IR24); Tetep and C101A51 | MABB | CAPS and STS marker | India | [177] |
| Vialone Nano | *Pita*, *Pib*, *Piz* | Blast resistance | Katy, Saber, Jefferson, Kusabue | MABB | dCAPS, InDel, and SNPs | Italia | [178] |
| Y58S, GuangZhan63S (GZ63), C815S, and HD9802S | *Pi37*, *Pit*, *Pid3*, *Pigm*, *Pi36*, *Pi5*, *Pi54*, *Pikm*, *Pb1* | Blast resistance | Q1333 (*Pi37*), K59 (*Pit*), Digu (*Pid3*), Gumei4 (*Pigm*), Q61 (*Pi36*), RIL260 (*Pi5*), Tsuyuake (*Pikm*), Tetep (*Pi54*) and Modan (*Pb1*) | MABB | SSR and Indel marker | China | [179] |
| Kinandang Patong, IRAT 109, Silewah, Basmati | *Gn1a*, *EFP* | Grain number 1a, Wealthy Farmer's Panicle | ST12 and ST6 | MABB | SSR | Japan | [180] |

**Table 2.** *Cont.*

| Recipient Genotype | QTL/Gene Transferred | Traits Transfer/Pyramiding | Donor Genotype | Method of Transfer | Marker Used | Country | Ref. |
|---|---|---|---|---|---|---|---|
| IR63307-4B-13-2 (indica), IRBB4/5/13/21 (indica), Kinandang Patong (japonica) | *pi21* | Blast resistance | Sensho | MABB | Indel marker | Philippines | [181] |
| | *Pib*, *Piz*, *Pik*, *Pita2*, *Piz-t* | Blast resistance | Saber, Kusabue, Katy, Jefferson, and Toride | MAS for pyramiding | | Italia | [182] |
| | *Bph14*, *Bph26*, *Bph17*, *Bph29* | BPH resistance | | MABB | | | [183] |
| Swarna-Sub1 | *qDTY1.1*, *qDTY2.1*, *qDTY3.1*, *Sub1* | Drought and submergence tolerant | N22/Swarna | MABB | SSR | Philippines, Nepal, India | [3] |
| Ningjing3 (japonica) and 93-11 (indica) | *Bph27(t)* | BPH resistance | Balamawe | MABB | Indel marker | China | [156] |
| Rassi | *Saltol* | Salt tolerance | FL478 (indica) | MABB | SSR | West Africa | [184] |
| WH421 (*xa4* and *xa21*) | *xa27*, *Pi9*, *Sub1A*, *Badh2.1* | Disease resistance, submergence tolerance and aromatic fragrance | KDML105, IRBB27, 75-1-127, IR64 (*Sub1ASub1A*), II-32A | MABB | SSR | China | [141] |
| Cultivar 9311 (indica) | *Pi9*, *Sub1A*, *xa21*, *xa27* | Blast resistance, blight resistance, submergence tolerance | 75-1-127 (*Pi9*), IR64 (*Sub1ASub1A*), WH21 (*xa21*), IRBB27 (*xa27*) | MABB | STS and SSR | China | [185] |
| Jinbubyeo (japonica) | *Pi40*, *xa4*, *xa5*, *xa21*, *Bph18* | Blast resistance, blight resistance, and BPH resistance | IRBB57 (*xa4+xa5+xa21*), IR65482-4-136-2-2 (*Pi40*), and IR65482-7-216-1-2 (*Bph18*) | MABB | SSR | Korea | [186] |
| Samba Mahsuri | *xa5*, *xa13*, and *Xa21* | Blight resistance | IRBB5 (*xa5*), IRBB13 (*xa13*), and IRBB21 (*xa21*) | MABB | STS marker | India | [187] |
| BRRI dhan49 | Saltol QTL | Salt tolerance | FL478 (indica) | MABB | SSR | Bangladesh | [188] |

Researchers must ensure that the introgression of the interest gene will not be lost during backcrossing by fixing the characteristics with the foreground selection (FGS) method [79,138,189–191]. FGS refers to the use of markers that are closely related to the interest gene to select target alleles/genes. Therefore, researchers have to find the marker that associates with the target characteristic. If a marker is available, the selection can be applied to each backcross generation to select plants that carry the specific allele (or gene, in the case of a new transgene).

The results of backcrosses must also be recovered from the genetic background of the recipient parent. To speed up this recovery, background selection (BGS) was carried out [79–81,143]. This selection refers to the use of markers that are not closely related to the gene of interest in selecting DNA other than that of the donor parent (in this case, selecting the allele of the recurrent parent at a locus other than the target locus). In BGS, donor parent alleles other than the target gene were intended to be eliminated (minimize introgression genes other than the interest gene). Early generations of backcrosses benefit the most from this strategy, although later generations may benefit from increased marker density.

Marker-assisted backcross selection (MABS) is principally a backcross method with the main target of inserting one or several genes or QTLs from donor parents to improve these characteristics in a superior cultivar that is advanced by utilizing genetic markers. This approach is the simplest form with the goal to incorporate a major gene from the donor parent with a resistant gene into a recurrent parent [192]. Utilization of MABS for selection in early generations of pyramidalization combined with phenotypic selection at the advanced stage is the most efficient strategy in inserting superior characteristics into one rice variety. Rice varieties that combine multiple environmental stress tolerances (multi-tolerant) can provide yield guarantees for farmers in areas where multiple stressors are frequently present in one growing season.

While the MABS is effective in transferring one or a few QTLs with strong genetic effects, for most complex characteristics, such as yield and biotic and tolerance/resistance characteristics, which are regulated by polygenes, small but additive genetic effects of QTLs, to combine all of the advantageous alleles in a single genotype can be challenging. In this case, the MARS strategy might be used as an alternative. MARS uses molecular markers to identify and select multiple genomic regions involved in the expression of complex traits in order to assemble the best-performing genotype within a single or across related populations [192]. MARS is one of the most important strategies in molecular breeding because it may help in the integration of multiple QTLs or favorable genes controlling the expression of a complex trait via recurrent selection based on multi-parent populations, which is a limitation of MABC [192].

This MARS approach captures many genomic areas to target a greater number of minor and major QTLs and accumulates a high number of QTLs in a given population utilizing a subset of specified markers that are significantly associated with target traits. As a consequence, the selection of genotype and intercrosses among the selected individuals can be made in the same crop season for one selection cycle. This can improve the efficiency of the selection process and hasten the integration of favorable genes from many parents into a single genotype [193].

IRRI researchers produced elite strains using Marker-Assisted Forward Breeding (MAFB) to introgress the resistance genes for four main diseases at once in rice: blast, bacterial leaf blight, brown planthopper, and gall midge into Swarna mega-cultivars containing drought tolerance genes [97]. Previously, researchers also used marker-assisted backcross breeding (MABB) to generate Nearly Isogenic lines (NILs) to introgress drought resistance genes from var. N22 into the Swarna-Sub1 mega-variety containing submerge resistance genes [144]. The same was also conducted by researchers in crossing var. Huazhan and var. H4 to produce F1 containing three QTLs carrying blast resistance genes, namely, *Pi1*, *Pi*46, and *Pita* [4]. The F1 was then used as the donor parent which was crossed with the R175 cultivar. An intercross F1 population was subsequently generated by crossing the F1 plants (Huazhan/H4) with R175. Das and Rao used the MABB method in assembling

multi-tolerant rice for resistance to bacterial blight, blast, gall midge, submergence, and resistance to salinity stress using improved fly cultivars as recurrent parents [5].

Pseudo-backcrossing as the modification of MABS is used to assemble multi-tolerant rice to combine genes for resistance to three major diseases (brown planthopper, leaf-neck bacterial blast, and leaf blight) [6] and flash flooding conditions in one aromatic rice cultivar, PinK3 [194]. Recently, researchers also used the pseudo-backcrossing method in assembling rice for multi-tolerance to drought, salinity, and submergence using the Improved White Ponni Variety [143].

While MAS can shorten breeding time, it is less suitable for quantitative traits governed by a large number of genes with small effects. To circumvent this constraint, genomic selection (GS) has been proposed as a viable option. Genomic selection is a subset of MAB in which each QTL is examined for linkage disequilibrium (LD) with at least one genetic marker [195]. This selection holds great potential to accelerate breeding progress and is cost-effective via early selection before phenotypes are measured [196], the ability to assess both additive and non-additive genetic variations accelerates the breeding cycle, making it easier to find superior genotypes quickly [197], and it is a possible strategy for improving complicated trait genetics while significantly reducing breeding cycles [195]. By incorporating machine learning-based predictive modelling in MAS, genomic selection achieves even more genetic gain while shortening crop life cycles. Genomic selection, which uses genetic information to predict an individual's potential performance for multiple traits, can be seamlessly integrated into the gene pyramiding process because it allows breeders to evaluate and select for multiple traits of interest at the same time, which aligns perfectly with the goals of gene pyramiding.

Models of GS can predict which trait combinations are most likely to produce superior variants. This enables breeders to make smart judgements on which characteristics to pyramid and which individuals to cross in order to attain the desired combinations, hence optimizing the entire breeding strategy. This guarantees that the desired features are efficiently selected. Because of the high number of single-nucleotide polymorphisms (SNPs) discovered by genome sequencing and the advent of new tools that effectively genotype large numbers of SNPs, this strategy is now more feasible. GS has been used in rice breeding, including inbred performance prediction, parental selection, and hybrid prediction [198,199]. GS is more successful in hybrid breeding because the genotypes of hybrids can be derived from their parents' genotypes rather than being sequenced from scratch, lowering the sequencing costs [200].

However, as a constraint, greater GS accuracy necessitates a lengthy development period and a high cost of SNP markers. Nowadays, genotyping-by-sequencing (GBS) technology is frequently employed in GS investigations to obtain high-density SNPs, but it requires a bioinformatic analysis approach and heavy imputation, as well as data sharing difficulties [201]. Nonetheless, technological advances such as high-throughput genotyping, phenotyping, genotype imputing, and sequencing technologies enable the generation of low-cost data. The worldwide rice community requires the development of an SNP array tailored for rice GS breeding [196]. Open-source systems for GS breeding have also been proposed to improve breeding efficiency [202].

Since the previous decade, modern plant breeding technologies such as MAS, NGS-based QTL mapping, genomic selection and editing, SNP chips, and high-throughput phenotyping have been used. Because of the development of association mapping, sequence-based mapping, and other methods for bringing out previously unknown variability in the natural gene pool, induced and inserted (activation tagging) mutagens may be able to contribute to the generation of novel variability [203–205]. Haplotype-based breeding is indeed a promising approach that is changing the landscape of gene pyramiding and crop improvement, particularly in the context of developing climate-smart rice varieties. Haplotype-based breeding allows breeders to combine alleles for distinct polymorphisms (such as SNPs, insertions/deletions, and other markers or variants) on the same chromosome that are inherited together with a low possibility of contemporaneous recombina-

tion [206]. In this method, specific genomic regions associated with desirable traits related to climate resilience in rice, such as drought tolerance, salt tolerance, heat resistance, disease resistance, and improved yield, are targeted [207–210].

Haplotype-related markers may be used to identify lines with unique recombination in chromosomal blocks of interest, allowing favorable and unfavorable genetic variation to be distinguished [210]. By identifying and introgressing favorable haplotypes from wild or donor rice varieties into elite cultivars, breeders can efficiently combine multiple beneficial traits [211]. Haplotypes enable more precise trait selection and introgression. Unlike traditional breeding, which may involve transferring entire chromosomal regions with linked undesirable traits, haplotype-based breeding allows for the selection of only the favorable alleles within the desired haplotype. This reduces the problem of linkage drag and ensures that only the desired traits are transferred. Haplotypes simplify the breeding process by focusing on specific genomic regions. This accelerates the development of climate-smart rice varieties as breeders can quickly identify and introgress the required haplotypes, reducing the number of generations required for trait fixation. Haplotypes derived from wild or landrace rice varieties often carry unique and valuable genetic traits related to climate resilience. The haplotypes of three tolerant genotypes, Goa Dhan 2, Panvel 1, and Goa wild rice (GWR) 005, appear to be completely different from the FL478 haplotype, indicating that tolerance in these genotypes is governed by a chromosomal area other than Saltol [211]. These three genotypes with likely unique regions for seedling-stage salt tolerance can be explored for improving rice cultivar salinity tolerance. By utilizing these haplotypes, breeders can introduce novel genetic diversity into cultivated rice varieties, enhancing their adaptability to changing environmental conditions [212]. Haplotype-based breeding enables the development of customized rice varieties tailored to specific agro-ecological zones and climate challenges. Researchers succeeded in identifying 120 previously functionally characterized key genes controlling grain yield (87 genes) and grain quality (33 genes) traits across the entire 3K RG pane in Nipponbare rice where all observed traits are affected by 21 genes [208]. This ensures that rice crops are better suited to local conditions, leading to increased productivity and sustainability. Haplotypes are identified through advanced genomic sequencing and data analysis. This data-driven approach empowers breeders with comprehensive genetic information, enabling them to make informed decisions and prioritize the most promising haplotypes for introgression.

Transcription Activator-Like Effector Nucleases (TALENs) and Clustered Regularly Interspaced Short Palindromic Repeats (CRISPRs/Cas9) have paved the way for genome editing that allows for the selective exchange of genes of interest [213,214]. Targeted gene editing for critical traits is becoming more popular because the products have no foreign material in their end product's DNA, so it can be defined as a non-GMO approach.

Editing LOX3 and acetolactate synthase (OsALS) genes with TALENs' application has succeeded in precisely improving herbicide resistance and seed storability in rice [215]. Similarly, CRISPR/Cas9 has been used to manipulate rice properties such as the yield component [216–219], herbicide resistance [220,221], explosion resistance [219,222–228], TGMS line development [229–231], stomata development [232,233], and amylose content modification [234–238].

## 4. Conclusions and Future Prospects

Ideotypes of superior rice continue to change from time to time according to the needs and demands of consumers. For this reason, breeders always try to meet these needs while at the same time ensuring that plant breeding activities are a never-ending program. Climate change is known to negatively affect rice productivity due to the emergence of various environmental stresses caused by changes in climate patterns that are different from usual. Plant breeding continues to provide a major contribution to anticipate food insecurity by allowing for large-scale food production with minimal resources while not affecting the ecological system. Rice will not experience a single stress as climate change worsens. Using resistant/tolerant rice cultivars may be one of the most effective strategies

to deal with abiotic and biotic pressures that is also affordable, durable, and ecologically beneficial. Breeders must consider how to address these issues by stacking multiple genes into a single rice genotype, known as gene pyramiding.

In the past decades, significant advancements in research and knowledge in rice tolerance have driven the increased application of the concept in its selection, especially with the application of the biotechnology approach, including marker-assisted selection to enhance, improve, and transform some processes. Technological developments in the field of genomics have enabled breeders to develop a molecular marker-assisted breeding technique that may be included in conventional breeding procedures rather than being thought of as a substitute. Genetic production may be increased further with the identification of QTLs through mapping, and the most beneficial ones were accumulated in the current high-yielding varieties using marker-assisted breeding. As a result of increased interest among rice geneticists worldwide in searching for more genes or QTLs from rice relatives (wild, weedy, and primitive cultivars), many prospective yield QTLs were discovered in *O. rufipogon*, *O. nivara*, and landrace.

When dealing with several features, gene pyramiding can be time-consuming and complex. In the near future, advanced technologies such as genomic prediction, gene editing, and allele mining have the potential to revolutionize and streamline the gene pyramiding process. The discovery of specific DNA markers linked to desired qualities is made possible by genomic prediction. Breeders can select individuals for mating based on their genetic makeup rather than observable qualities if they have a deep understanding of the organism's genome. This enables more accurate and efficient trait selection.

Genomic prediction aids in determining the precise locations of genes responsible for various phenotypes. This data helps breeders discover and track the inheritance of several genes at the same time, making it easier to combine desired features in a single organism. Genomic prediction models make use of massive amounts of genomic data to forecast an individual's probable trait performance. This enables breeders to make more educated judgements regarding which individuals to cross, improving the likelihood of gene pyramiding success.

CRISPR-Cas9 gene editing, on the other hand, allows for precise changes in an organism's DNA. This allows breeders to directly introduce or alter specific genes linked to desirable traits, bypassing generations of traditional breeding. Gene editing technologies with multiplexing capabilities can target and modify many genes at the same time. This greatly speeds up the process of introducing several features into a single organism, making gene pyramiding more effective.

Allele mining (genome sequencing and data mining, introgressing of novel alleles) is another advancement in technology that may facilitate the process of gene pyramiding in the near future. Because of advances in genome sequencing and bioinformatics technologies, it is now possible to investigate the genetic variety within a species. Breeders can uncover unusual or previously unknown alleles associated with desired qualities by mining the genetic data of numerous individuals. Breeders can use allele mining to find and introduce novel alleles from wild or underutilized genetic resources into cultivated types. This broadens the genetic pool and expands the possibilities for gene pyramiding.

Although technological innovation has contributed significantly towards the development of climate-smart rice, the imperative need is to ascertain strategic plans for the development of varieties possessing multiple stress tolerances. Future studies should also focus on a clearer understanding of the mechanism of plants related to each stressor that still needs to be explored to sharpen the activities of assembling the intended variety.

Furthermore, disseminating CSR product results to farmers is a significant challenge. As a result, to maximize benefits, it is necessary to develop a research strategy that can be implemented directly with farmers. The selection of activities that involve direct collaboration with farmers should be prioritized in order to increase dissemination and ensure food security, particularly in climatically susceptible agro-ecologies, for future generations.

**Author Contributions:** Conceptualization, I.I. and N.C.; writing—original draft preparation, I.I.; writing—review and editing, I.I., Y.N., N.B. and N.C.; visualization, I.I. and Y.N.; supervision, N.C. All authors have read and agreed to the published version of the manuscript.

**Funding:** This research received no external funding.

**Institutional Review Board Statement:** Not applicable.

**Informed Consent Statement:** Not applicable.

**Data Availability Statement:** Not applicable.

**Acknowledgments:** Financial assistance from the Directorate General of Higher Education and the Education Financing Service Center of the Ministry of Education, Cultural, Research and Technology is greatly appreciated.

**Conflicts of Interest:** The authors declare no conflict of interest.

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
