# Peer review of "Toward Food Security in 2050: Gene Pyramiding for Climate-Smart Rice"

_sustainability, doi:10.3390/su151914253_

Round 1

Reviewer 1 Report

Properly written manuscript,  Tables and diagrams are clear and easy to understand for the readers and relevant to the topic.  Some minor suggestions in the manuscript need to be fixed, which I have listed in the author section, and a few suggestions for improving the manuscript are necessary at some points.

Line no. 49 “This" is because the nature of tolerance to abiotic stresses is inherited in a complex manner and phenotype selection method which need large breeding material population and time consume that make also slows down the selection for these characters” (this sentence is not clear look at the highlighted part and correct it)

Line no.69 “We have discussed related various studies of new technique breeding methods for a stacked multi-tolerance gene to abiotic stress into one superior rice variety” (correct this sentence)

Line no 39,41,124,145 please rectify grammatical mistakes in the sentences from these lines

 Line no. 131, “It h" reported that the first rice variety produced in Indonesia was var. Begawan in 131 1943. The BengaBengawan's rice has a genetic background of varieties from China, var. 132 Latisail from India, and var. Benong from Indonesia” (Is "it Begawan or Bengawan?  make it clear.)

 Line no. 189 “Afte" an extensive safety assessment, the Philippines will become the first country in the world to authorize Golden Rice for commercial multiplication in 2021” (correct the sentence)

 Line no. 332 “Rice" breeding at many research centres has developed climate change ready rice that can withstand unfavourable environments that occur more frequently and intensively due to climate change.” (mention some examples of climate-ready rice/ climate-smart rice varieties developed by breeding with proper reference)

 Line no. 508identification of QTL this stress tolerance in the early and vegetative growth phases of rice and candidate genes using the genotyping-by-sequencing (GBS) approach to compose SNP markers that will be used for genotype selection and map preparation” (correct this sentence)

In Tabe 1, “Table 1. Several rice QTLs for tolerance to environmental problems have been successfully identified using a marker-assisted breeding technique.” QTL" can be identified using the marker-assisted selection (MAS) technique, not with marker-assisted breeding, modify it)

The introduction section looks short; please extend it by adding information related to some developed climate-smart crops (rice and other crops) and introgression methods

  In the sub-sub-section' te-smart rice,the authors have only focused on climate change and their effects on the crops, so add some detail about rice varieties with improved traits. Please provide examples of developed rice cultivars resistant to different biotic and abiotic stress. Give detailed information about approaches adapted for their development and mention success stories  Give proper numbering in all subtitles

Line no 39,41,124,145 please rectify grammatical mistakes in the sentences from these lines

Author Response

Dear Reviewer,

Thank you very much for taking your valuable time to review our manuscript.

We have revised the manuscript to reflect most of the suggestions provided by the reviewers. We have highlighted the changes within the manuscript.

Herewith we provide our responses.

Comments and suggestions

Responses

Line no. 49 “This" is because the nature of tolerance to abiotic stresses is inherited in a complex manner and phenotype selection method which need large breeding material population and time consume that make also slows down the selection for these characters” (this sentence is not clear look at the highlighted part and correct it).

Thank you for pointing this out, we have revised the sentence at Line 49.

Line no.69 “We have discussed related various studies of new technique breeding methods for a stacked multi-tolerance gene to abiotic stress into one superior rice variety” (correct this sentence).

Done. Line 70.

Line no 39,41,124,145 please rectify grammatical mistakes in the sentences from these lines.

Done. Line 38-39, Line 41, Line 122, Line 143

Line no. 131, “It h" reported that the first rice variety produced in Indonesia was var. Begawan in 131 1943. The BengaBengawan's rice has a genetic background of varieties from China, var. 132 Latisail from India, and var. Benong from Indonesia” (Is "it Begawan or Bengawan?  make it clear.).

The correct one is var. Begawan. The correct sentence at Line 130.

 Line no. 189 “Afte" an extensive safety assessment, the Philippines will become the first country in the world to authorize Golden Rice for commercial multiplication in 2021” (correct the sentence).

Done. The correct sentence at Line 188:

After an extensive safety assessment, the Philippines became the first country in the world to authorize Golden Rice for commercial multiplication in 2021 [26].

 Line no. 332 “Rice" breeding at many research centres has developed climate change ready rice that can withstand unfavourable environments that occur more frequently and intensively due to climate change.” (mention some examples of climate-ready rice/ climate-smart rice varieties developed by breeding with proper reference).

Thank you for pointing this out, we have revised the sentence at Line 331.

Line no. 508 “identification of QTL this stress tolerance in the early and vegetative growth phases of rice and candidate genes using the genotyping-by-sequencing (GBS) approach to compose SNP markers that will be used for genotype selection and map preparation” (correct this sentence).

Done. Line 544.

In Tabe 1, “Table 1. Several rice QTLs for tolerance to environmental problems have been successfully identified using a marker-assisted breeding technique.” QTL" can be identified using the marker-assisted selection (MAS) technique, not with marker-assisted breeding, modify it)

Thank you for pointing this out, we have revised the sentence at Line 499.

In the sub-sub-section' the-smart rice, the authors have only focused on climate change and their effects on the crops, so add some detail about rice varieties with improved traits. Please provide examples of developed rice cultivars resistant to different biotic and abiotic stress. Give detailed information about approaches adapted for their development and mention success stories  

Thank you for the suggestion. Anyway, we have shown this in Table 2 (Line 570).

Give proper numbering in all subtitles.

Thanks for the correction. We have corrected it according to the instructions and can be seen on Line 32, 75, 105, 106, 268, 367 and 732.

Line no 39,41,124,145 please rectify grammatical mistakes in the sentences from these lines.

Done. Line 38-39, Line 41, Line 122 and Line 143.

Reviewer 2 Report

The manuscript entitled “Toward Food Security 2050: Gene Pyramiding for Climate-2 Smart Rice” was reviewed thoroughly. The manuscript covers extensive information on gene pyramiding and its role in food security. Authors have made great efforts to compile huge information available on gene identification and pyramiding in rice breeding. Some of the suggestions to improve the manuscript are provided hereunder-

Some information on the impact of gene pyramided varieties in mitigating climate vagaries in terms of a percent increased production and monetary benefits.

It is a review article, headings like methodology, results, etc. do not have any relevance.

A table of the most commonly introgressed gene/QTL should be provided.

There are plentiful advances in genomic prediction technology, authors should add a paragraph on advances in the integration of genomic selection with gene pyramiding.

How advanced technologies like genomic prediction, gene editing, and allele mining may ease the process of gene pyramiding in the near future? Through some light on this to improve the impact of the manuscript.

Haplotype identification and haplotype-based introgression are changing the process of gene pyramiding. Through some light on haplotype-based breeding and its role in developing climate-smart rice varieties.

The concept of QTL hot spots and their importance in gene pyramiding may be included in the manuscript.

On an overall basis, the manuscript is written well and accommodates a large amount of information on gene/QTL identification and introgression to develop climate-smart rice varieties. The manuscript is of great interest to rice breeders and researchers across the globe. 

Minor English language editing by a native English speaker is required.

Author Response

Dear Reviewer,

Thank you very much for taking your valuable time to review our manuscript.

We do appreciate the time and effort that you and the reviewers have dedicated to providing valuable feedback on our manuscript. We have revised the manuscript to reflect most of the suggestions provided by the reviewers. Herewith, the appendix table at the bottom of the email details the improvements we've made point by point. We have highlighted the changes within the manuscript.

In addition to the above comments, all spelling and grammatical errors have been corrected. We also would have academic proofread from the editorial team. We look forward to hearing from you in due time regarding our submission and to respond to any further questions and comments you may have.

Herewith our response below.

Thank you.

Comments and suggestions

Responses

Some information on the impact of gene pyramided varieties in mitigating climate vagaries in terms of a percent increased production and monetary benefits.

Thank you for the suggestion. Examples of gene pyramidization applications that have seen their impact are the improvements of several well-known varieties that we have shown, especially in the section regarding the STRASA project (Line 221).

The STRASA project was a big success. Until the end of the project in 2019, more than 150 varieties tolerant to flood, drought, and salinity, including multiple tolerance improvements in mega rice varieties, Swarna, Sambha Mahsuri, Sahbhagi dhan, were released in South Asia and in sub-Saharan Africa. STRASA also contacted 35 million farmers, produced more than 1 million tons of seeds, with 18 million hectares of covered area

It is a review article, headings like methodology, results, etc. do not have any relevance.

Sustainability now accepts free format submission.

A table of the most commonly introgressed gene/QTL should be provided

List of rice QTLs that were successfully identified is shown in Table 1 (Line 499), and success stories of rice QTL introgression are shown in Table 2 (Line 570).

There are plentiful advances in genomic prediction technology, authors should add a paragraph on advances in the integration of genomic selection with gene pyramiding. 

Thank you for pointing this out, we have add some paragraphs to emphasis this point on start from Line 642:

While MAS can shorten breeding time, it is less suitable for quantitative traits governed by a large number of genes with small effects. To circumvent this constraint, genomic selection (GS) has been proposed as a viable option. Genomic selection is a subset of MAB in which each QTL is examined for linkage disequilibrium (LD) with at least one genetic marker (Sinha et al, 2023). This selection holds a great potential to accelerate breeding progress and is cost-effective via early selection before phenotypes are measured (Xu et al, 2021),  ability to assess both additive and non-additive genetic variations accelerates the breeding cycle, making it easier to find superior genotypes quickly (Crosssa et al 2017), and is a possible strategy for improving complicated trait genetics while significantly reducing breeding cycles (Sinha et al 2023). By incorporating machine-learning-based predictive modelling in MAS, genomic selection achieves even more genetic gain while shortening crop life cycles. Genomic selection, which uses genetic information to predict an individual's potential performance for multiple traits, can be seamlessly integrated into the gene pyramiding process because it allows breeders to evaluate and select for multiple traits of interest at the same time, which aligns perfectly with the goals of gene pyramiding.

Models of GS can predict which trait combinations are most likely to produce superior variants. This enables breeders to make smart judgements on which characteristics to pyramid and which individuals to cross in order to attain the desired combinations, hence optimising the entire breeding strategy. This guarantees that the desired features are efficiently selected. Because of the high number of single nucleotide polymorphisms (SNPs) discovered by genome sequencing and the advent of new tools that effectively genotype large numbers of SNPs, this strategy is now more feasible. GS has been used in rice breeding, including inbred performance prediction, parental selection, and hybrid prediction (Wang et al, 2017, Xu et al, 2014). GS is more successful in hybrid breeding because the genotypes of hybrids can be derived from their parents’ genotypes rather than being sequenced from scratch, lowering the sequencing costs (Xu et al, 2018).

However, as a constraint, greater GS accuracy necessitates a lengthy development period and a high cost of SNP markers. Nowadays, genotyping-by-sequencing (GBS) technology is frequently employed in GS investigations to get high-density SNPs, but it requires a bioinformatic analysis approach and heavy imputation, as well as data sharing difficulties (Chen et al, 2014). Nonetheless, technological advances such as high-throughput genotyping, phenotyping, genotype imputing, and sequencing technologies enable the generation of low-cost data. The worldwide rice community requires the development of an SNP array tailored for rice GS breeding (Xu et al, 2021). Open-source systems for GS breeding have also been proposed to improve breeding efficiency (Spindel and McCouch, 2016).

How advanced technologies like genomic prediction, gene editing, and allele mining may ease the process of gene pyramiding in the near future? Through some light on this to improve the impact of the manuscript.

Thank you for pointing this out, we have revised to emphasis this point on Subtitle 4. Conclusion and Future Prospect, start from line 757:

When dealing with several features, gene pyramiding can be time-consuming and complex. In the near future, advanced technologies such as genomic prediction, gene editing, and allele mining have the potential to revolutionize and streamline the gene pyramiding process. The discovery of specific DNA markers linked to desired qualities is made possible by genomic prediction. Breeders can select individuals for mating based on their genetic makeup rather than observable qualities if they have a deep understanding of the organism’s genome. This enables more accurate and efficient trait selection.

Genomic prediction aids in determining the precise locations of genes responsible for various phenotypes. This data helps breeders discover and track the inheritance of several genes at the same time, making it easier to combine desired features in a single organism. Genomic prediction models make use of massive amounts of genomic data to forecast an individual's probable trait performance. This enables breeders to make more educated judgements regarding which individuals to cross, improving the likelihood of gene pyramiding success.

CRISPR-Cas9 gene editing, on the other hand, allows for precise changes in an organism's DNA. This allows breeders to directly introduce or alter specific genes linked to desirable traits, bypassing generations of traditional breeding. Gene editing technologies with multiplexing capabilities can target and modify many genes at the same time. This greatly speeds up the process of introducing several features into a single organism, making gene pyramiding more effective.

Allele mining (genome sequencing and data mining, introgressing of novel alleles) is another advancement technology that may facilitate the process of gene pyramiding in the near future. Because of advances in genome sequencing and bioinformatics technologies, it is now possible to investigate the genetic variety within a species. Breeders can uncover unusual or previously unknown alleles associated with desired qualities by mining the genetic data of numerous individuals. Breeders can use allele mining to find and introduce novel alleles from wild or underutilised genetic resources into cultivated types. This broadens the genetic pool and expands the possibilities for gene pyramiding.

Haplotype identification and haplotype-based introgression are changing the process of gene pyramiding. Through some light on haplotype-based breeding and its role in developing climate-smart rice varieties. 

Thank you for pointing this out, we have revised to emphasis this point on start from Line 682:

Haplotype-based breeding is indeed a promising approach that is changing the landscape of gene pyramiding and crop improvement, particularly in the context of developing climate-smart rice varieties. Haplotype-based breeding allows breeders to combine alleles for distinct polymorphisms (such as SNPs, insertions/deletions, and other markers or variants) on the same chromosome that are inherited together with a low possibility of contemporaneous recombination (Bhat et al, 2021). In this method, specific genomic regions associated with desirable traits related to climate resilience in rice, such as drought tolerance, salt tolerance; heat resistance, disease resistance, and improved yield are targeted (Wang et al 2023; Abbai et al 2019; Li et al, 2022; Mei et al 2022; Zhu et al, 2016; Qian et al, 2017).

Haplotype-related markers may be used to identify lines with unique recombination in chromosomal blocks of interest, allowing favourable and unfavourable genetic variation to be distinguished (Qian et al, 2017). By identifying and introgressing favorable haplotypes from wild or donor rice varieties into elite cultivars, breeders can efficiently combine multiple beneficial traits (Manohara 2019). Haplotypes enable more precise trait selection and introgression. Unlike traditional breeding, which may involve transferring entire chromosomal regions with linked undesirable traits, haplotype-based breeding allows for the selection of only the favorable alleles within the desired haplotype. This reduces the problem of linkage drag and ensures that only the desired traits are transferred. Haplotypes simplify the breeding process by focusing on specific genomic regions. This accelerates the development of climate-smart rice varieties as breeders can quickly identify and introgress the required haplotypes, reducing the number of generations required for trait fixation. Haplotypes derived from wild or landrace rice varieties often carry unique and valuable genetic traits related to climate resilience. The haplotypes of three tolerant genotypes, Goa Dhan 2, Panvel 1, and Goa wild rice (GWR) 005, appear to be completely different from the FL478 haplotype, indicating that tolerance in these genotypes is governed by a chromosomal area other than Saltol (Manohara et al 2021). These three genotypes with likely unique regions for seedling stage salt tolerance can be explored for improving rice cultivar salinity tolerance. By utilizing these haplotypes, breeders can introduce novel genetic diversity into cultivated rice varieties, enhancing their adaptability to changing environmental conditions (Maung et al 2021). Haplotype-based breeding enables the development of customized rice varieties tailored to specific agroecological zones and climate challenges. Researchers succeeded in identifying 120 previously functionally characterized key genes controlling grain yield (87 genes) and grain quality (33 genes) traits across the entire 3K RG pane in Nipponbare rice (Abbai et al 2019). Plant height, number of tillers, days to flowering, panicle length, primary branch numbers per panicle, grain yield, grain size, grain amylose content, grain iron and zinc concentration are all affected by 21 genes (Abbai et al 2019). This ensures that rice crops are better suited to local conditions, leading to increased productivity and sustainability. Haplotypes are identified through advanced genomic sequencing and data analysis. This data-driven approach empowers breeders with comprehensive genetic information, enabling them to make informed decisions and prioritize the most promising haplotypes for introgression.

The concept of QTL hot spots and their importance in gene pyramiding may be included in the manuscript. 

We have add some paragraph to emphasis this point on start from Line 512:

QTL (Quantitative Trait Locus) hot spots are specific regions within a genome where multiple QTLs associated with different desirable traits are clustered or co-located. In other words, these are regions of genome where multiple genes or genetic markers that influence various quantitative traits of interest are found close together. The Swarna*3/Morobekerjaan derived population was studied in order to identify six genomic areas for early vigour and related traits, two of which were QTL hotspots (QTL hotspot A and QTL hotspot B), which harboured practically for early vigor and related traits under dry direct-seeded system (Singh et al 2017). Satrio et al (2023) detected the presence of a QTL hotspot that was strongly associated with drought tolerance on rice chromosome 8 in the RIL population derived from rice IR64 × Hawara Bunar using differential gene expression meta-analysis and qRT-PCR technique.

QTL hot spots are of great importance in the context of gene pyramiding, which is the process of stacking multiple desirable traits into a single crop variety. QTL hot spots make it easier to combine multiple desirable traits in a single breeding program. Since several QTLs are concentrated in one genomic region, breeders can target that region to introgress multiple traits simultaneously, reducing the number of generations and resources needed for pyramiding. QTL hot spots can help minimize linkage drag because the multiple traits of interest are close together, allowing breeders to select for the specific QTLs they want while avoiding those associated with undesired traits. By leveraging QTL hot spots, breeders can develop improved crop varieties more rapidly. This is particularly valuable when breeding for complex traits, such as disease resistance, where multiple genes may be involved. Hot spots allow breeders to identify a genomic region associated with resistance to multiple diseases, thereby streamlining the breeding process. QTL hot spots enable more precise breeding strategies. Breeders can use marker-assisted selection (MAS) or gene editing techniques to specifically target the QTLs within the hot spot, ensuring that the desired traits are incorporated into the final variety without unnecessary genetic baggage. Stacking multiple QTLs from a hot spot can lead to synergistic effects, resulting in crop varieties with enhanced performance. Combining QTLs for high yield and disease resistance from a hot spot, for example, can yield varieties with both traits, providing a more productive and resilient crop.

On an overall basis, the manuscript is written well and accommodates a large amount of information on gene/QTL identification and introgression to develop climate-smart rice varieties. The manuscript is of great interest to rice breeders and researchers across the globe. 

Thank you very much.
